# Constitutively active RAS prolongs Cdc42 signalling, while MAPK signalling is attenuated during fission yeast mating

Emma J. Kelsall[1], Akatsuki Kimura[2,3,4]*, Ábel Vértesy[5¤a], Kornelis R. Straatman[6], Mishal Tariq[1], Raquel Gadea[1,7], Chandni Parmar[1], Gabriele Schreiber[5], Shubhchintan Randhawa[1¤b], Takashi Y. Ida[8], Cyril Dominguez[1,9], Edda Klipp[5]*, Kayoko Tanaka[1]*

**1** Division of Molecular and Cell Biology, School of Biological and Biomedical Sciences, University of Leicester, Leicester, United Kingdom, **2** Center for Data Assimilation Research and Applications, Joint Support-Center for Data Science Research, Research Organization of Information and Systems (ROIS), Tachikawa, Japan, **3** Cell Architecture Laboratory, National Institute of Genetics, Mishima, Japan, **4** Genetics Program, The Graduate University for Advanced Studies, Sokendai, Mishima, Japan, **5** Theoretical Biophysics, Institute of Biology, Humboldt-Universität zu Berlin, Berlin, Germany, **6** Centre for Core Biotechnology Services, University of Leicester, Leicester, United Kingdom, **7** Marco Formacion Profesional, Zaragoza, Spain, **8** Faculty of Science, Nara Women's University, Nara, Japan, **9** Leicester Institute of Structural and Chemical Biology, University of Leicester, Leicester, United Kingdom

☯ These authors contributed equally and are listed alphabetically according to their family names.
¤a Present address: Institute of Molecular Biotechnology of the Austrian Academy of Sciences (IMBA), Vienna, Austria
¤b Present address: QPharma Inc., Morristown, New Jersey, United States of America
* kt96@le.ac.uk (KT); edda.klipp@rz.hu-berlin.de (EK); akkimura@nig.ac.jp (AK)

## Abstract

The small GTPase RAS is a signalling hub activating multiple pathways, which may respond differently to a constitutively active RAS mutation. We explored this issue in fission yeast, where RAS-mediated pheromone signalling (PS) activates two downstream pathways: the MAPK[Spk1] and Cdc42 pathways. We observed that the yeast RAS mutation *ras1.G17V*, an equivalent of the mammalian *ras.G12V* oncogenic mutation, causes prolonged Cdc42 activation, whereas MAPK[Spk1] activation was transient and attenuated. To explain this observation, we generated a PS framework by conducting genetic epistasis analysis of PS mutants and biochemical analysis of two Ras1 effectors, Cdc42-GEF[Scd1] and MAPKKK[Byr2], each of which triggers activation of the Cdc42 and MAPK[Spk1] pathways, respectively. Cdc42-GEF[Scd1] and MAPKKK[Byr2] directly interacted with Ras1 *in vitro* in a competitive manner, and overexpression of the Ras binding domain of either Cdc42-GEF[Scd1] or MAPKKK[Byr2] in cells inhibited both downstream pathways, confirming that Ras1 signalling branches into the MAPK[Spk1] and Cdc42 pathways. In conjunction with the genetic epistasis analysis, we developed the PS framework-based mathematical model to test which network structures can explain the transient MAPK[Spk1] activation profile. Incorporating a negative-feedback circuit acting on pheromone production or sensing enabled the model to quantitatively reproduce MAPK[Spk1] dynamics in the wild type and 20 additional PS

**Data availability statement:** The codes used in this study are available on our GitHub (https://github.com/akkimura/pombePheromoneSignalling2025_250423f).

**Funding:** This study was supported by BBSRC BB/S019510/1 to KT and KRS, by Wellcome Trust Institutional Strategic Support Fund WT097828/Z/11/Z and WT097828/Z/11/B to KT, by the Deutsche Forschungsgemeinschaft (DFG) EXC81 to EK, by the Joint Support-Center for Data Science Research, ROIS and JSPS KAKENHI JP18H02414 to AK, and by the German Academic Exchange Service (DAAD) A0981674 to AV. The funders had no role in study design, data collection and analysis, decision to publish, or preparation of the manuscript.

**Competing interests:** The authors have declared that no competing interests exist.

mutants. The predicted PS negative-feedback was experimentally confirmed by deleting Sxa2, the carboxypeptidase that degrades one of the mating pheromones, which led to hyperactivation of both MAPK[Spk1] and Cdc42. Our study provides a holistic understanding of the fission yeast pheromone signalling network, explaining how RAS signalling propagates differently through two downstream pathways. Our PS mathematical model may serve as a valuable reference framework for analysing other RAS signalling systems.

## Author summary

As a signalling hub central to cell proliferation and differentiation, constitutively active RAS mutations in humans cause profound effects, including cancer and developmental disorders. However, whether these mutations lead to widespread activation of all the downstream pathways remains elusive. We addressed this question by employing a highly tractable model system, fission yeast, in which pheromone signalling involves RAS, which activates two pathways, MAPK and Cdc42. We observed that a constitutively active RAS mutation prolongs Cdc42 activation, whereas MAPK activation was transient and attenuated. To explain this observation, we constructed a yeast pheromone signalling framework using genetic and biochemical approaches. We applied mathematical modelling to refine the framework further. The model predicted that negative feedback that acts on pheromone production or sensing plays a key role in MAPK signal attenuation. We tested this prediction experimentally by genetically increasing pheromone production, which led to increased MAPK activation, thereby validating it.

Our work demonstrates that constitutively active RAS signalling can impact downstream pathways differently, and that modulation of a signalling unit upstream of RAS can influence the outcome of a constitutively active RAS mutation. Our mathematical model may serve as a valuable tool for analysing other RAS signalling systems.

## Introduction

Proto-oncogene Ras GTPase family members are widely conserved and play pivotal roles in cell growth, differentiation and apoptosis [1]. The physiological impact of Ras mutations is highlighted in the resultant tumorigenesis and developmental disorders [2,3]. More than 99% of identified oncogenic RAS mutations occur at codons 12, 13 and 61 of human Ras isoforms [2] and impair efficient GTP hydrolysis and/or enhance the GDP-GTP nucleotide exchange activity [4,5]. This results in the accumulation of GTP-bound Ras, and when overexpressed, the oncogenic Ras leads to constitutive activation of the downstream effector pathways, such as the ERK signalling pathway [6,7]. Interestingly, however, previous studies using mouse embryonic

fibroblasts (MEFs) demonstrated that oncogenic Ras expressed at its endogenous level does not cause ERK pathway hyperactivation upon a growth factor stimulation [8–10], although the *KRAS^(G12D)* MEFs showed enhanced proliferation, morphological changes and partial transformation [9]. We also observed the lack of hyperactivation of the ERK pathway in a human cell culture model that carries heterozygous oncogenic G12X mutations despite increased cell proliferation and motility [11]. Meanwhile, small GTPases, including RalA/B, Cdc42 and Rac, are required in oncogenic-RAS-driven tumorigenesis [12–18] and the oncogenic RAS may cause small GTPase hyperactivation. However, whether the lack of ERK hyper-activation and an enhanced small GTPase activation are a conserved fundamental feature of the oncogenic RAS signalling is unclear. We wished to address the question using the model organism fission yeast, where a unique Ras homologue, Ras1, plays a crucial role in pheromone signalling by activating two downstream pathways, a pheromone MAPK cascade and Cdc42 pathway, leading to the mating of haploid cells [19,20] (Fig 1A and 1B).

Upon nutritional starvation, fission yeast cells of opposite mating types (*h^+* and *h^-*) exchange mating pheromones [19]. Gpa1, the α-subunit of the pheromone receptor-coupled G-protein, relays the pheromone signal into the cell [21] through Ras1, leading to the activation of the MAPK cascade consisting of Byr2 (MAPKKK), Byr1 (MAPKK) and Spk1 (MAPK) [22–28]. The *ras1.G17V* mutant, an equivalent of mammalian oncogenic *RAS.G12V* mutation, produces an excessively elongated shmoo, or a conjugation tube, upon exposure to the mating pheromone [24], implying that Ras1.G17V may amplify the pheromone signal [19].

Ras1 also regulates cell morphology during vegetative growth: whilst deletion of either *gpa1*, *MAPKKK^(byr2)*, *MAPKK^(byr1)* or *MAPK^(spk1)* does not result in any obvious phenotypes during vegetative cell growth [21,29,30], *ras1Δ* cells lose the typical rod-shaped morphology to become rounded [24,31]. Studies based on recombinant protein assays and yeast-2-hybrid analysis identified Scd1, a GDP-GTP exchange factor (GEF) for Cdc42, which regulates the actin cytoskeleton and cell morphology, as well as MAPKKK^(Byr2), as Ras1 interacting proteins [32–34]. These signalling components were all found in the cell cortex before mating [20,35–38]. These observations suggest that Ras1 regulates the pheromone MAPK^(Spk1) and the Cdc42 pathways. However, how a constitutive activation of Ras1 affects the two downstream pathways has yet to be understood.

By establishing conditions that induce highly synchronous mating of fission yeast cells, we were able to quantify the MAPK^(Spk1) and Cdc42 activation dynamics during the physiological mating process in wildtype and in mutants of various mating phenotypes. We conducted genetic epistasis analysis of the mating-deficient mutants and biochemical analysis of Cdc42-GEF^(Scd1) and MAPKKK^(Byr2) to generate a framework of the fission yeast pheromone signalling. Based on the signalling framework, we built a mathematical model, which was optimised using the experimental data of MAPK^(Spk1) and Cdc42 activation dynamics of the wildtype and mating-deficient mutants. The model predicted a negative feedback circuit acting on the pheromone production or sensing, which we experimentally confirmed. The model serves as a prototype of a branched Ras-mediated signalling pathway, demonstrating that the *ras1.G17V* mutation does not induce constitutive activation of the MAPK^(Spk1) but rather prolongs Cdc42 activation. The model also highlights the physiological importance of the bipartite activation of MAPKKK^(Byr2): a Ras1-dependent and a Ras1-independent mechanism, the latter of which employs the adaptor protein Ste4 [39,40]. The crucial role of Cdc42 in the *ras1.G17V* mutant in causing the *ras1.G17V* phenotype was also noted.

## Results

### A highly synchronous mating assay allows quantifying the MAPK^(Spk1) activity during the fission yeast mating process

To monitor fission yeast pheromone signalling throughout the mating process (Fig 1A and 1B), we established a protocol to induce highly synchronous mating, employing cells where the endogenous MAPK^(Spk1) is tagged with GFP-2xFLAG and conducted quantitative Western blotting of phosphorylated MAPK^(Spk1) (S1 Fig). Under this condition, homothallic *h^(90)* cells started to mate 6 hours after induction of mating (Fig 1C, grey line). The phosphorylated (active) MAPK^(Spk1) (**pp**MAPK^(Spk1))

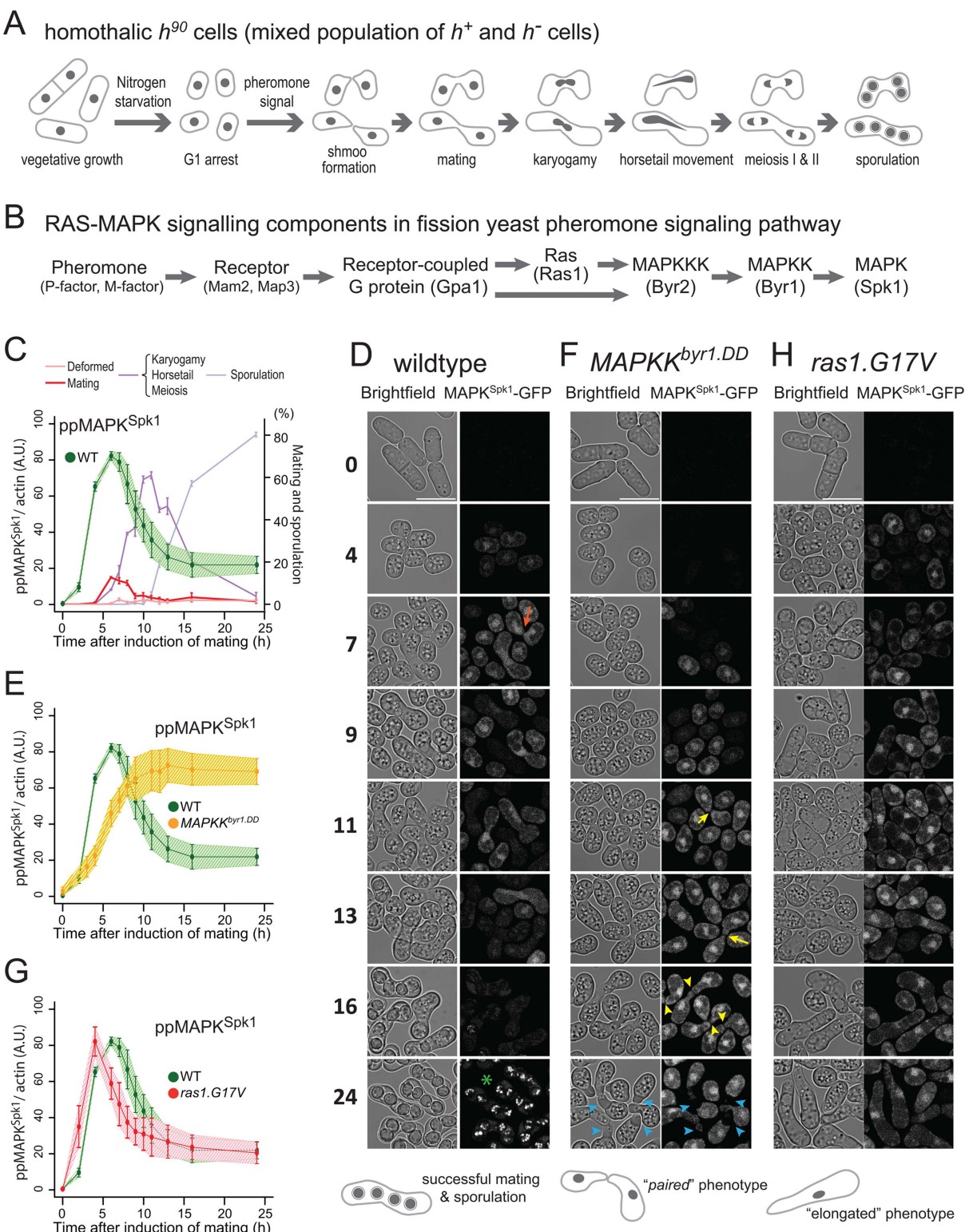

**Fig 1. Distinct modes of MAPK$^{Spk1}$ temporal phosphorylation profile and morphological changes during sexual differentiation in wildtype, *MAPKK$^{byr1.DD}$* and *ras1.G17V* mutants. (A)** A pictorial representation of wildtype fission yeast sexual differentiation. **(B)** A list of key signalling components of the fission yeast pheromone signalling pathway. The diagram reflects the prediction that Gpa1 and Ras1 separately contribute to MAPKKK$^{Byr2}$

activation, although the precise mechanism is unknown (Xu et al., 1994). At the same time, Ras1 activation is expected to be at least partly under the influence of active Gpa1 because the *ste6* gene, encoding a Ras1 activator, is strongly induced upon successful pheromone signalling (Hughes et al., 1994). **(C)-(H)** Cells were induced for sexual differentiation by the plate mating assay system as described in the materials and methods. **(C)**, **(E)** and **(G)** Quantified **pp**MAPK^Spk1 signal from western blots of wildtype (KT3082) **(C)**, *MAPKK*byr1.DD (KT3435) **(E)** and *ras1.G17V* (KT3084) **(G)** cells. Three biological replicates, shown in S2 Fig, were used for quantitation (error bars are ± SD. Individual data is shown in S3 Fig). Actin was used as a loading control, and quantitation was carried out using the Image Studio ver2.1 software (Licor Odyssey CLx Scanner). For the wildtype samples in **(C)**, the % ratios of cells at the indicated sexual differentiation stages are also indicated. In this counting, each zygote was counted as two cells to reflect the mating status of the whole population (n = 400, three biological replicates). The wildtype **pp**MAPK^Spk1 result **(C)** is also presented in **(E)** and **(G)** as a reference. **(D)**, **(F)** and **(H)** Cellular morphology (brightfield) and localisation of MAPK^Spk1-GFP over a 24-hour time-course in wildtype **(D)**, *MAPKK*byr1.DD **(F)** and *ras1.G17V* **(H)** cells. The time after induction of mating in hours is indicated on the left. At each time point, a brightfield image and a GFP signal image were taken as described in Materials and Methods. For each image, a single Z image of brightfield and a corresponding GFP image are presented. A green asterisk in the time 24 h in the wildtype cell image **(D)** indicates auto-fluorescence signals that accumulate in mature spores. An orange arrow in panel **(D)** indicates a pair of mating cells. Yellow arrows and arrowheads in panel **(F)** indicate cells forming a pair, but fail to proceed to cell fusion, exhibiting the "fus" or "paired" phenotype. Blue arrowheads indicate vacuoles developing at the conjugation tubes of the "paired" cells. Note that orange and yellow arrows/arrowheads also indicate accumulation of MAPK^Spk1-GFP at the shmoo tips. Scale bars represent 10μm.

signal was first detected three hours after induction of mating and reached its peak at about five to seven hours when cell fusion was also initially observed (Fig 1C, green line. Original membrane images and quantitations presented in S2 and S3 Figs). The **pp**MAPK^Spk1 then gradually decreased to a non-zero level as meiosis continued towards sporulation. It was also noted that the total MAPK^Spk1-GFP was essentially not expressed during the vegetative cycle, but it was promptly induced by nitrogen starvation (S2 and S3 Figs). The *mapk^spk1* gene is a known target of the transcription factor Ste11 [41], which itself is activated by MAPK^Spk1 [42]. This positive feedback loop likely facilitates a swift increase of MAPK^Spk1 expression upon nitrogen starvation.

We found that MAPK^Spk1-GFP localised to the cytosol, cell cortex and the nucleus, with some nuclear accumulation, before gradually disappearing as the mating process came to an end (Figs 1D, S4 and S5). The quantification of the MAPK^Spk1-GFP in wildtype cells undergoing the sexual differentiation stages demonstrated both the nuclear and total GFP signal per cell/zygote peaked at the time of mating (for haploid cells) and karyogamy (for diploid zygotes) (S4 Fig), consistent with the Western blotting result. Transient foci of GFP signals were also found at the cell cortex, especially at the shmoo tips, as has been reported for MAPKK^Byr1, the activator of MAPK^Spk1 [36] (Figs 1D, orange arrow, S4 and S5).

As there is a strong correlation between the timings of peak MAPK^Spk1 activation and mating/cell fusion, we examined whether cell fusion is prerequisite for MAPK^Spk1 downregulation, utilising cells deficient in *fus1* gene, which encodes a yeast formin homologue that plays an essential role in the cell fusion event during the mating process by regulating the cytoskeletal and membrane organisation [43–48]. In the *fus1Δ* mutant cells, the increased ppMAPK^Spk1 level stayed high 12 hours after induction of sexual differentiation, and the cells accumulated the MAPK^Spk1-GFP particularly in the nucleus and the shmoo tips (S6 Fig). The result demonstrated that successful mating is required to downregulate the MAPK^Spk1.

## Constitutively active MAPKK^Byr1.DD causes constitutive activation of MAPK^Spk1

Activation of MAPKK family kinases is mediated by dual phosphorylation of conserved Ser/Thr residues [49], and when the corresponding Ser214/Thr218 were substituted to aspartic acid, the resultant fission yeast MAPKK^Byr1.DD was expected to act as a constitutively active MAPKK [50]. Indeed, the level of **pp**MAPK^Spk1 in the *MAPKK^byr1.DD* mutant remained high after reaching its highest intensity at around 13 hours after induction (Figs 1E, yellow line, S2 and S3), although the initial increase of the **pp**MAPK^Spk1 signal was slower compared to the wildtype strain. The **pp**MAPK^Spk1 level remained high even 48 hours after the induction of mating (S7 Fig). The result highlights that the endogenously expressed MAPKK^Byr1.DD is proficient in retaining the high **pp**MAPK^Spk1 level, and the counteracting de-phosphorylation by phosphatases Pmp1 and Pyp1 [51] is not efficient in downregulating the **pp**MAPK^Spk1 signal in the presence of MAPKK^Byr1.DD. Consistent with the observed slower increase in **pp**MAPK^Spk1, the nuclear localisation of MAPK^Spk1 was also delayed compared to the wildtype cells (Fig 1F). A strong nuclear MAPK^Spk1-GFP signal was then observed in the "paired" fusion-deficient cells, an intriguing

phenotype of the *MAPKK*<sup>byr1.DD</sup> cells (Fig 1F) [36, 50]. Interestingly, the projection tips of the paring cells often show increased MAPK<sup>Spk1</sup>-GFP signal (Figs 1F, yellow arrows and S8).

## The *ras1.G17V* mutation causes immediate but transient MAPK<sup>Spk1</sup> activation

The fission yeast equivalent of human oncogenic *RAS.G12V* is *ras1.G17V,* which induces an excessively elongated shmoo, and the cells fail to recognize a partner and become sterile [22,24]. The "elongated shmoo" phenotype was interpreted as an excess activation of the Ras1 downstream pathway(s), leading to a prediction that the *ras1.G17V* causes over-activation of MAPK<sup>Spk1</sup> [19,20]. However, the **pp**MAPK<sup>Spk1</sup> signal intensity declined comparably to that in wildtype cells, indicating that down-regulation of **pp**MAPK<sup>Spk1</sup> is effective in the *ras1.G17V* mutant, unlike in the *MAPKK*<sup>byr1.DD</sup> mutant (Figs 1G, red line, S2 and S3). Correspondingly, the nuclear, cytoplasm and cell cortex MAPK<sup>Spk1</sup>-GFP signals also declined 16–24 hours after induction of mating (Figs 1H and S8). Interestingly, an immediate increase of the **pp**MAPK<sup>Spk1</sup> signal upon induction of mating was observed (Fig 1G, red line).

## The elongated *ras1.G17V* shmoos develop with a detectable level of MAPK<sup>Spk1</sup>: neither amplitude nor duration of ppMAPK<sup>Spk1</sup> signal influences the *ras1.G17V* morphological phenotype

Having observed that *MAPKK*<sup>byr1.DD</sup> and *ras1.G17V* show different MAPK<sup>Spk1</sup> activation profiles (sustained vs transient) and different morphological phenotypes ("paired" vs elongated), we examined a correlation between the MAPK<sup>Spk1</sup> activation profiles and the cell morphology. The *ras1.G17V MAPKK*<sup>byr1.DD</sup> double mutant showed constitutive activation of MAPK<sup>Spk1</sup> (Figs 2A, light blue line, S2 and S3). The nuclear **pp**MAPK<sup>Spk1</sup>-GFP signal in the *ras1.G17V MAPKK*<sup>byr1.DD</sup> double mutant was present 24 hours after induction of mating, unlike the *ras1.G17V* single mutant cells (Fig 2B), confirming that the *ras1. G17V MAPKK*<sup>byr1.DD</sup> double mutant cells retained a high **pp**MAPK<sup>Spk1</sup> level. Thus, *MAPKK*<sup>byr1.DD</sup> is overall epistatic to *ras1. G17V* in terms of the MAPK<sup>Spk1</sup> activation status.

However, for cell morphology, the *ras1.G17V MAPKK*<sup>byr1.DD</sup> double mutant cells showed the "elongated" *ras1.G17V* phenotype (Fig 2B), demonstrating that *ras1.G17V* was epistatic to *MAPKK*<sup>byr1.DD</sup> regarding cell morphology.

Interestingly, the morphological change in the *ras1.G17V MAPKK*<sup>byr1.DD</sup> double mutant was first noticed 12 hours after induction of mating, much later than the *ras1.G17V* single mutant, but as a comparable timing as the "pair" formation of the *MAPKK*<sup>byr1.DD</sup> single mutant (Fig 2B). Given the slower increase of the **pp**MAPK<sup>Spk1</sup> signal in the *MAPKK*<sup>byr1.DD</sup> mutant, we predicted that the "elongated" shmoo formation still requires a certain level of MAPK<sup>Spk1</sup> activity. Indeed, when *MAP-KK*<sup>byr1</sup> was deleted in the *ras1.G17V* mutant, not only was the MAPK<sup>Spk1</sup> activation abolished (S1G Fig), but also shmoo formation was abolished (Fig 2C). Based on these observations, we concluded that the Ras1 status, but not the MAPK<sup>Spk1</sup> activation profile, determines the cell morphology. Yet, the *ras1.G17V* "elongated shmoo" phenotype still requires a certain level of MAPK<sup>Spk1</sup> activity, which defines the timing of the shmoo formation.

## Cdc42 is required for the shmoo formation and full activation of the MAPK<sup>Spk1</sup>

During the vegetative cycle, *ras1Δ* cells show spherical cell morphology [22,24] where polarised localisation of active Cdc42 is compromised [52,53] (S9 Fig) which can be detected by CRIB-GFP, a specific binder of the active GTP-bound form of Cdc42 [54]. The current understanding is that Ras1 activates Cdc42, which leads to the activation of downstream Ste20-like kinase, Pak1/Shk1 [55–58], resulting in actin reorganisation and shmoo formation under mating conditions [35,59].

The elongated shmoo phenotype was lost when the *scd1*, encoding a Cdc42-GEF, was deleted in the *ras1.G17V* mutant. Instead, the cells showed a mating-deficient phenotype similar to the *cdc42-GEF*<sup>scd1</sup>Δ single mutant, supporting the model that Cdc42 acts downstream of Ras1 to cause morphological changes (Fig 3A).

Interestingly, the **pp**MAPK<sup>Spk1</sup> level was substantially reduced in the *cdc42-GEF*<sup>scd1</sup>Δ mutant compared to the wildtype (Figs 3B, purple line, S2 and S3). The result agreed with a previous study that predicted active Cdc42 to contribute to the

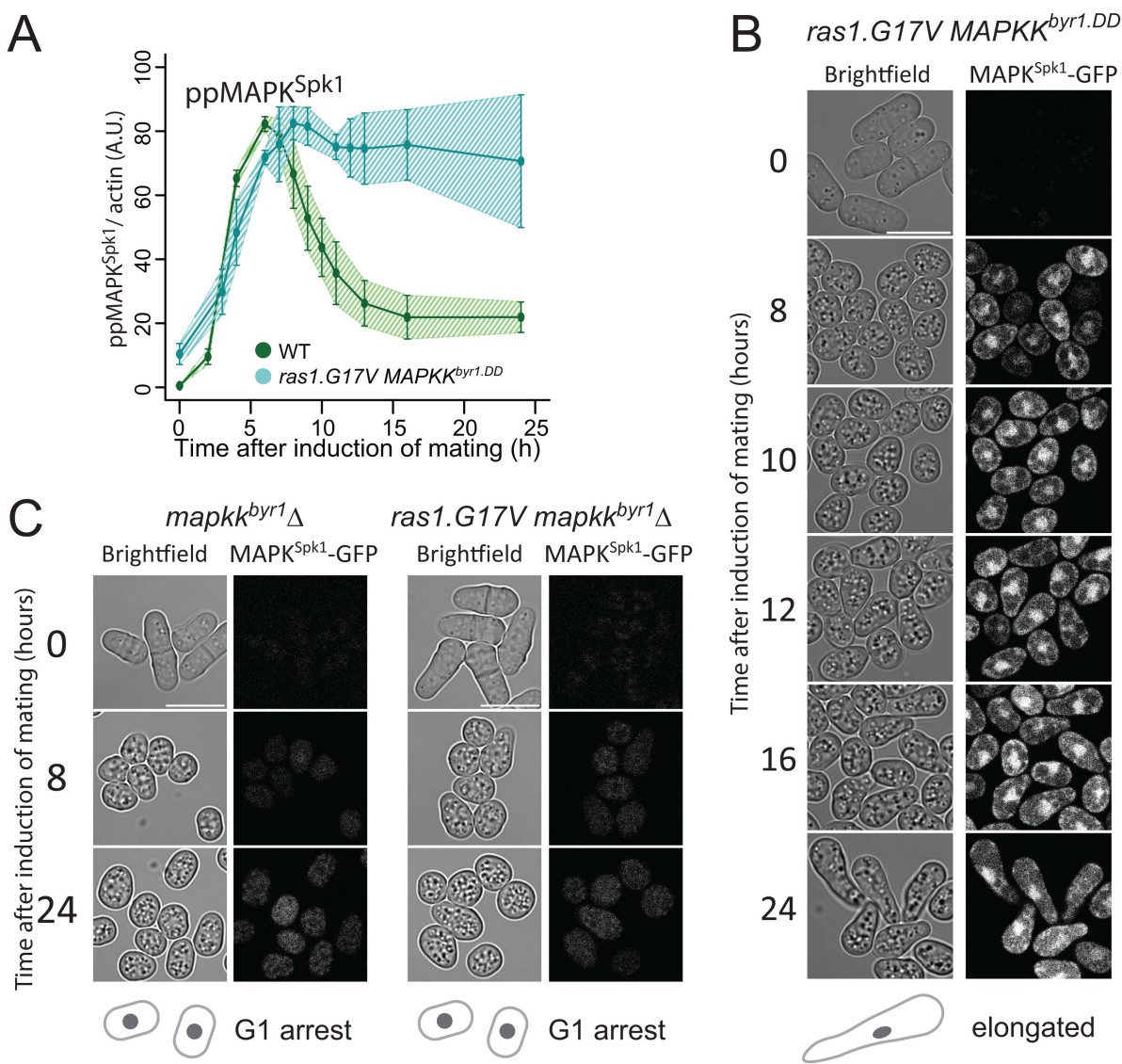

**Fig 2. In the *ras1.G17V MAPKK^byr1.DD* double mutant, the MAPK^Spk1 phosphorylation profile follows *MAPKK^byr1.DD* single mutant phenotype whilst cell morphology mimics the *ras1.G17V* single mutant phenotype. (A)** MAPK^Spk1 phosphorylation status in the *ras1.G17V MAPKK*byr1.DD double mutant cells (KT3439). Cells were induced for mating by the plate mating assay system as described in the materials and methods. Quantitated **pp**MAPK^Spk1 signal (arbitrary unit) from western blots is presented. Original membrane images are presented in S2 Fig. Three biological replicates were used for quantitation (error bars are ± SD). α-tubulin was used as a loading control, and quantitation was carried out using the Image Studio ver2.1 software (Licor Odyssey CLx Scanner). The wildtype **pp**MAPK^Spk1 result (Fig 1C) is also presented as a reference. **(B)** The terminal mating phenotype of *ras1.G17V MAPKK*byr1.DD double mutant (KT3439) is a phenocopy of *ras1.G17V* single mutant which shows the "elongated" morphology. Images were taken of *ras1.G17V MAPKK*byr1.DD double mutant (KT3439) in the same way as in Fig 1. Time after induction of mating in hours is indicated on the left. **(C)** There is no morphological change in the absence of MAPK^Spk1 signalling. Cell images of *MAPKK*byr1Δ (KT4300) and *ras1.G17V MAPKK*byr1Δ (KT5215) strains are shown. Images were taken in the same way as in Fig 1. Time after induction of mating in hours is indicated on the left of each series. The scale bars represent 10μm.

activation of MAPKKK^Byr2 [33]. Even so, some MAPK^Spk1 activation occurred in the *cdc42-GEF^scd1*Δ mutant, and a nuclear MAPK^Spk1-GFP signal was low, but detectable (Fig 3A).

A reduced but substantial level of MAPK^Spk1 activation in the *cdc42-GEF^scd1*Δ mutant means that the mating deficiency of this mutant is unlikely to result from the lack of MAPK^Spk1 activation. Indeed, the introduction of *MAPKK^byr1.DD*

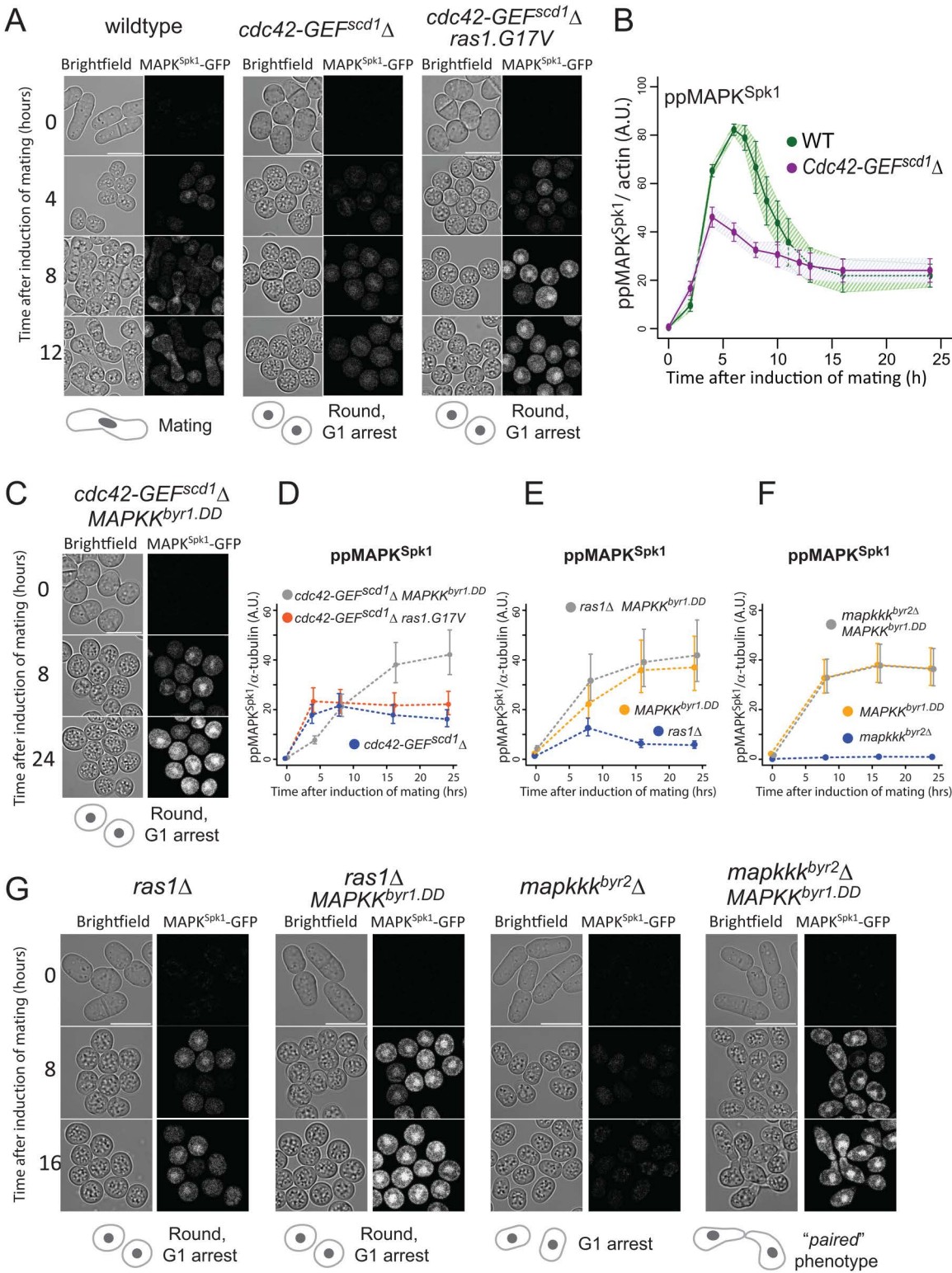

**Fig 3. Ras1 activates both MAPK$^{Spk1}$ and Cdc42 pathways during pheromone signalling. (A)** The deletion of *scd1* causes round cell morphology and sterility. Images of WT (KT3082), *scd1Δ* (KT4061) and *scd1Δ ras1.G17V* double mutant (KT4056), expressing the MAPK$^{Spk1}$-GFP, were taken in the same way as in Fig 1. Numbers on the left represents hours after induction of mating. **(B)** The **pp**MAPK$^{Spk1}$ levels during the sexual differentiation

in *scd1Δ* (KT4061) cells. Results of three biological replicates (error bars are±SD) are presented. Original membrane images are presented in S2 Fig. The wildtype **pp**MAPK$^{Spk1}$ result presented in Fig 1C is also shown in green as a reference. **(C)** Cell images of *scd1Δ MAPKK*byr1.DD double mutant (KT4047). The images were taken in the same way as in Fig 1. Numbers on the left represent hours after induction of mating. **(D)** The **pp**MAPK$^{Spk1}$ levels in *scd1Δ* (KT4061), *scd1Δ MAPKK*byr1.DD (KT4047) and *scd1Δ ras1.G17V* (KT4056) cell extracts. Original Western blotting data is presented in S10A Fig and the total MAPK$^{Spk1}$ levels in these samples are presented in S10D Fig. **(E)** The **pp**MAPK$^{Spk1}$ levels in *ras1Δ* (KT4323), *MAPKK*byr1.DD (KT3435) and *ras1Δ MAPKK*byr1.DD (KT4359) cell extracts. Original Western blotting data is presented in S10B Fig and the total MAPK$^{Spk1}$ levels in these samples are presented in S10E Fig. **(F)** The **pp**MAPK$^{Spk1}$ levels in *mapkkk*byr2Δ (KT3763), *MAPKK*byr1.DD (KT3435) and *mapkkk*byr2Δ *MAPKK*byr1.DD (KT4010) cell extracts. Original Western blotting data is presented in S10C Fig and the total MAPK$^{Spk1}$ levels in these samples are presented in S10F Fig. For **(D)**, **(E)** and **(F)**, quantification was carried out using the Image Studio ver2.1 (Li-cor). **(G)** Cell images of the strains mentioned in **(E)** and **(F)** were taken in the same way as in Fig 1. Numbers on the left represent hours after induction of mating. For all the images presented in **(A)**, **(C)** and **(G)**, the scale bars represent 10μm.

to the *cdc42-GEF$^{scd1}$Δ* mutant did not restore the mating deficient phenotype, even though nuclear MAPK$^{Spk1}$-GFP highly accumulated and the ppMAPK$^{Spk1}$-GFP level was substantially increased (Figs 3C, 3D, and S10). The result shows that Cdc42 activity is required for the mating process regardless of the MAPK$^{Spk1}$ activation status. Taken together with the essential role of MAPK$^{Spk1}$, we concluded that the mating pheromone signalling feeds into two pathways, MAPK$^{Spk1}$ and Cdc42.

## Ras1 activates two effector pathways, MAPK$^{Spk1}$ and Cdc42

In order to further clarify the role of Ras1, we examined the MAPK$^{Spk1}$ activation status and cell morphology in the following four strains: *ras1Δ* mutant, *ras1Δ MAPKK$^{byr1.DD}$* double mutant, *MAPKKK$^{byr2}$Δ* mutant and *MAPKKK$^{byr2}$Δ MAPKK$^{byr1.DD}$* double mutant. As mentioned earlier, the vegetatively growing *ras1Δ* cells show a round morphology with reduced cortical signal of CRIB-GFP, demonstrating that Cdc42 activation is compromised (S9 Fig). The deletion of *ras1* also causes substantial reduction, but not complete elimination, of the **pp**MAPK$^{Spk1}$ (Figs 3E, blue line, and S10B); thus, Ras1 plays an important role in activating both Cdc42 and MAPK$^{Spk1}$ pathways. Introduction of the *MAPKK$^{byr1.DD}$* mutation into the *ras1Δ* mutant cells induces the constitutive **pp**MAPK$^{Spk1}$ (Figs 3E, grey line and S10B) but does not affect the round cell morphology, and cells remain sterile (Fig 3G, the 2$^{nd}$ left panel).

   In striking contrast, the sterile phenotype of the *MAPKKK$^{byr2}$Δ*, associated with a complete lack of shmoo formation (Fig 3G, the 2$^{nd}$ right panel), was converted to the "*paired*" phenotype when combined with the *MAPKK$^{byr1.DD}$* mutation (Fig 3G, the far right panel). As expected, the *MAPKKK$^{byr2}$Δ MAPKK$^{byr1.DD}$* double mutant shows MAPK$^{Spk1}$ constitutive activation (Figs 3F and S10C). Thus, unlike the cases of *scd1Δ* or *ras1Δ*, the lack of *MAPKKK$^{byr2}$* can be bypassed by constitutive activation of MAPK$^{Spk1}$, indicating that the sole role of MAPKKK$^{Byr2}$ is to activate the MAPK$^{Spk1}$, unlike its upstream activator, Ras1, which also activates Cdc42 pathway.

   It was also noted that the loss of MAPKKK$^{Byr2}$ completely blocked the MAPK$^{Spk1}$ production, whereas deletion of *scd1* did not cause detectable changes to the total MAPK$^{Spk1}$ level and the *ras1Δ* cells still produced a detectable level of MAPK$^{Spk1}$ (S10D, S10E and S10F Fig).

## Ras1.G17V causes accumulation of Cdc42-GTP at the cell cortex

Having observed a relatively mild influence of Ras1.G17V towards the MAPK$^{Spk1}$ activation profile, we next examined whether the Cdc42 pathway was affected by the *ras1.G17V* mutation. As previously observed, dynamic foci of CRIB-GFP appeared on the cell cortex upon induction of mating [59] (Fig 4A and 4B). In our experimental condition, more than 80% of the wildtype cells showed the cortical CRIB-GFP signal at 4.5 hours after induction of mating (Fig 4B). The cortical CRIB-GFP foci became concentrated at the mating site and quickly disappeared once cells fused successfully to form zygotes (Fig 4A and 4B). In striking contrast, in the *ras1.G17V* mutant cells, the cortical CRIB-GFP signal persisted, often at the elongated tip end of the cells, even 12.5 hours after induction of mating (Fig 4A and 4B). The signal could still be

**A** ★ Cells with cortical foci of CRIB-GFP (an indicator of Cdc42$^{GTP}$) after induction of mating

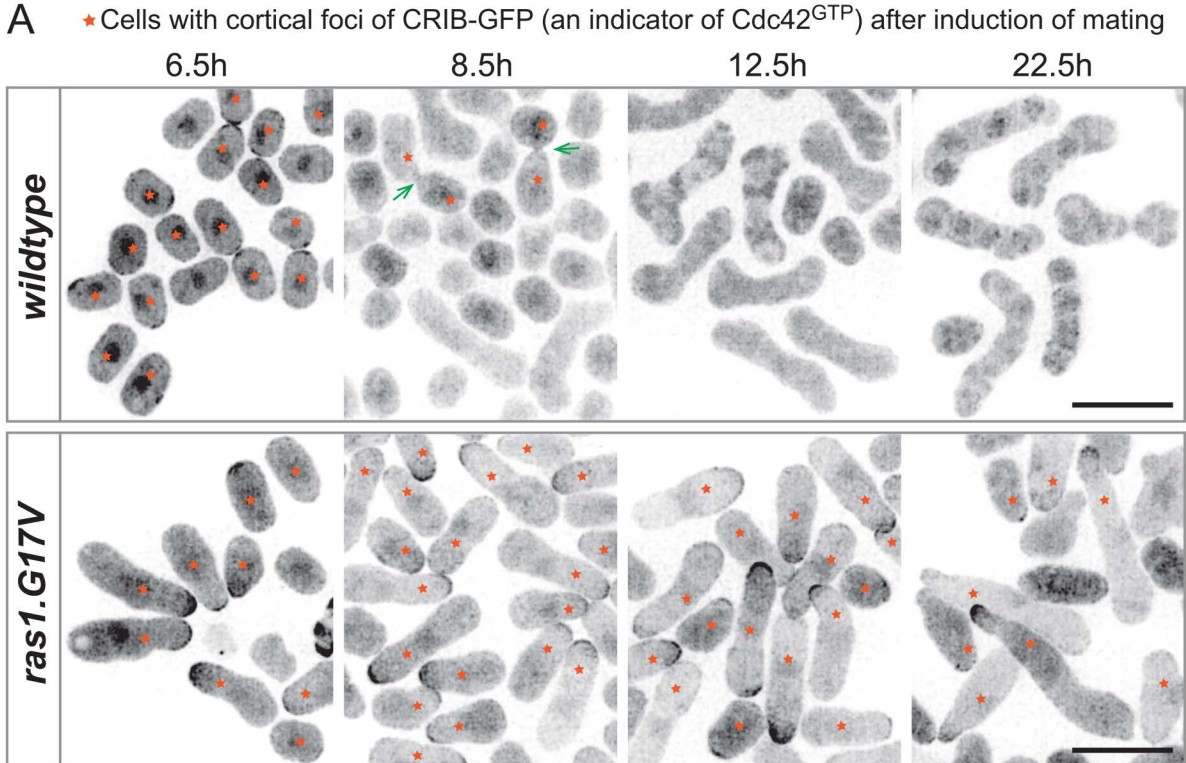

**B** % Cells with cortical CRIB-GFP foci upon induction of mating

**Fig 4. Ras1.G17V induces cortical Cdc42$^{GTP}$ accumulation. (A)** Cell morphology and localisation of Cdc42$^{GTP}$, indicated by CRIB-GFP signal, during the sexual differentiation process. Wildtype (KT5077) and *ras1.G17V* (KT5082) mutant cells were induced for sexual differentiation by the plate mating assay condition (Materials and Methods), and live cell images were taken at the indicated time after the induction. Representative CRIB-GFP signal images are presented. Orange stars indicate cells with cortical CRIB-GFP foci. Rapidly disappearing CRIB-GFP signals at the fusion site of wildtype mating cells are indicated by green arrows at the time 8.5h image. The scale bar represents 10 μm. **(B)** Quantification of the results presented in **(A)**. At each time point (4.5h, 6.5h, 10.5h, 12.5h and 22.5h after induction of sexual differentiation), 150 cells were examined to determine whether they have cortical CRIB-GFP foci. % cells with cortical CRIB-GFP foci are presented. The experiment was repeated three times, and the mean values and SDs were plotted in the graph.

seen in about 40% of the cells 22.5 hours after induction of mating (Fig 4A and 4B). The result shows that the Cdc42 pathway is excessively activated in the *ras1.G17V* mutant, and the tip localisation of Cdc42$^{GTP}$ indicates that the signature "elongated" *ras1.G17V* morphological phenotype is caused by hyperactivation of the Cdc42 pathway.

The effect of *ras1.G17V* mutation towards Cdc42 activation was also observed during vegetative growth, producing the strongest cortical CRIB-GFP signal at the cell tip in the *ras1.G17V* mutant (S9 Fig), as previously reported [20]. The effect of *ras1. G17V* on cell morphology was best recognised when the *rga4* that encodes a GTPase activation protein negatively regulating Cdc42 [52,54,60] was deleted, as the resultant *rga4Δ ras1.G17V* double mutant cells were larger and round with an increased CRIB-GFP signal (S9 Fig). These results fit the hypothesis that Ras1.G17V has an enhanced capability to activate Cdc42.

## Two Ras effectors, MAPKKK$^{Byr2}$ and Cdc42-GEF$^{Scd1}$, compete with each other for Ras1

We next examined whether the two Ras effectors, MAPKKK$^{Byr2}$ and Cdc42-GEF$^{Scd1}$, compete with each other for Ras1. First, we overexpressed the Ras binding domains (RBDs) of MAPKKK$^{Byr2}$ and Cdc42-GEF$^{Scd1}$ in the *ras1.G17V* cells and examined the activation status of MAPK$^{Spk1}$ and Cdc42. For MAPKKK$^{Byr2}$, the region spanning the residues 65–180 (Byr2-RBD) was used following previous structural studies [61], and for Cdc42-GEF$^{Scd1}$, the region spanning the residues 760–872 (Scd1-PB1) was used based on its amino acid sequence similarity with the Ras Associating (RA) domain of mammalian RalGDS.

Overexpression of Byr2-RBD in the *ras1.G17V* cells substantially reduced the **pp**MAPK$^{Spk1}$ level, whereas the total MAPK$^{Spk1}$ level was only marginally affected (Figs 5A and S11), indicating that the Byr2-RBD competed against the endogenous full-length MAPKKK$^{Byr2}$ in activating the MAPK$^{Spk1}$. Strikingly, overexpression of Scd1-PB1 also inhibited the MAPK-$^{Spk1}$ activation, exhibiting its capability to interfere with the MAPK$^{Spk1}$ pathway (Figs 5A and S11).

When Cdc42 activation was examined in the *ras1.G17V* cells, overexpression of Scd1-PB1 reduced the cortical CRIB-GFP signal and abolished deformed cells (Fig 5B and 5C), indicating that the endogenous Scd1 function was interfered with the over-expressed Scd1-PB1. Byr2-RBD overexpression also resulted in a comparable phenotype (Fig 5B and 5C). Collectively, two Ras1 effectors, MAPKKK$^{Byr2}$ and Cdc42-GEF$^{Scd1}$, are capable of competing with each other for Ras1 *in vivo* when overexpressed. The effect of overexpression of Scd1-PB1 and Byr2-RBD on cell morphology was also observed in vegetatively growing cells, consistent with the Ras1 function of maintaining the cell morphology of vegetative cells (S12 Fig).

Next, we conducted *in vitro* binding competition assays using bacterially expressed GST-tagged fragments of Byr2-RBD and Scd1-PB1, both of which bound Ras1.G17V (1–172) loaded with GTP (Fig 5D). The binding of Ras1.G17V$^{GTP}$ to GST-Byr2-RBD was substantially decreased when the GST-Scd1-PB1 was added (Fig 5E). The result indicates the biochemical competitive nature of MAPKKK$^{Byr2}$ and Cdc42-GEF$^{Scd1}$ for active Ras1.

We also probed the *in vivo* status of MAPKKK$^{Byr2}$ and Cdc42-GEF$^{Scd1}$ during the mating process by tagging these proteins with a 1xGFP tag, which may produce minimal artefact compared to a more sensitive 3xGFP. Scd1 was tagged at its C-terminal end, and MAPKKK$^{Byr2}$ was tagged at its N-terminal end, as C-terminally tagged MAPKKK$^{Byr2}$ cells became sterile (S13A Fig). In vegetatively growing cells, the Scd1-GFP signal was detected at the growing cell tips as previously described [62], whilst the GFP-Byr2 signal was undetectable (S13B Fig). In cells undergoing sexual differentiation, we could detect both Scd1-GFP and GFP-Byr2 signals at the shmoo tip of mating cells (S13C Fig). As both signal intensities were low, rigorous quantification could not be performed. However, both signals were within a similar intensity range, and it is unlikely that one is in excess of the other; hence, a competition between MAPKKK$^{Byr2}$ and Cdc42-GEF$^{Scd1}$ might exist, provided that the number of available Ras1 molecules is limited.

## Ras1 and an adaptor protein Ste4 are both necessary to fully activate MAPKKK$^{Byr2}$

Although Ras1 plays a significant role in activating MAPK$^{Spk1}$, a low but detectable level of **pp**MAPK$^{Spk1}$ was still induced in the *ras1Δ* mutant (Figs 3E and S10B), indicating that there is a Ras1-independent mechanism to activate MAPK$^{Spk1}$.

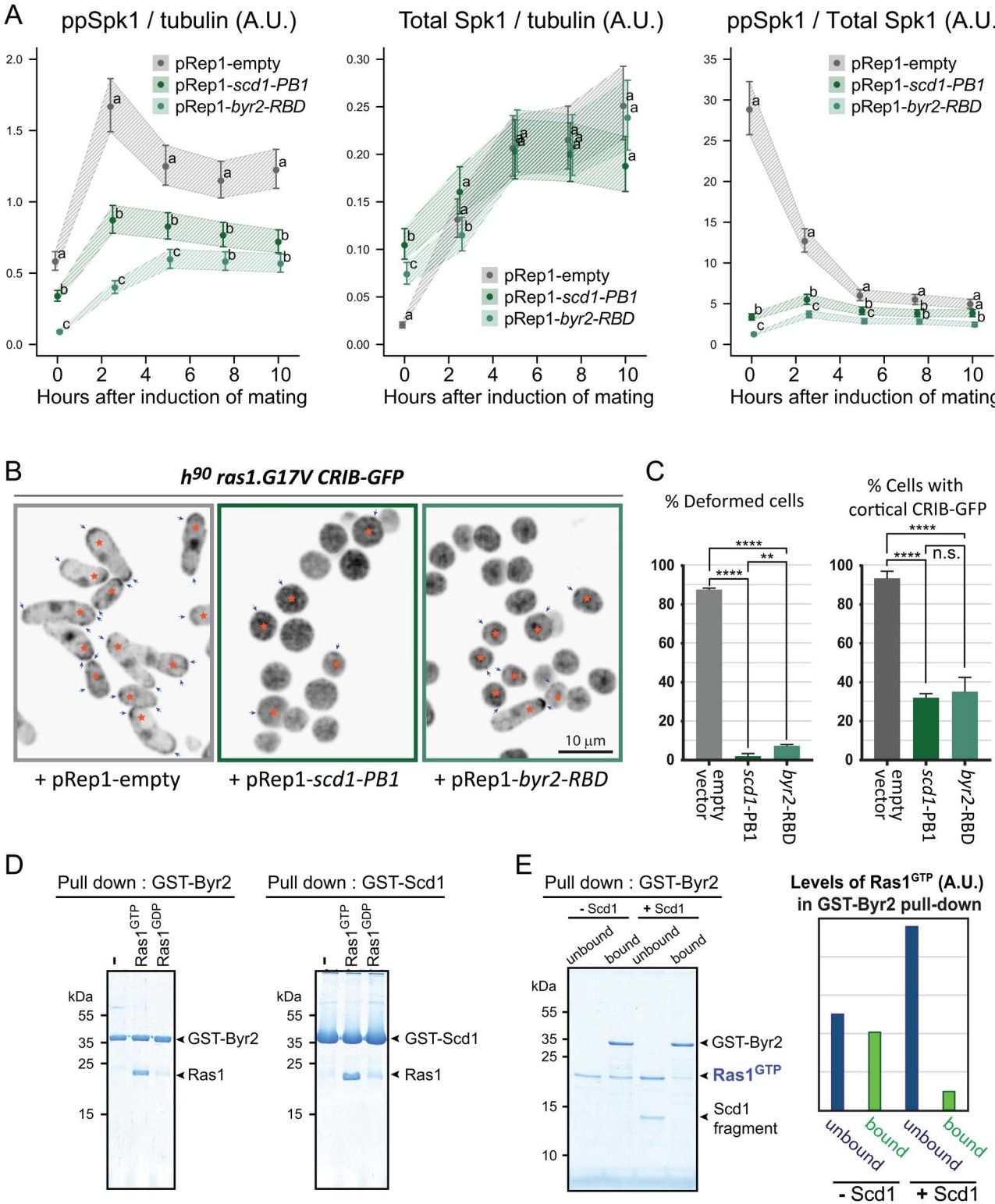

**Fig 5. Two Ras effectors, MAPKKK<sup>Byr2</sup> and Cdc42-GEF<sup>Scd1</sup>, compete each other for Ras1. (A)** Cells harboring *ras1.G17V* and MAPK<sup>Spk1</sup>-GFP-2xFLAG (KT5940) were transformed with either pRep1 empty vector, pRep1-*scd1*(760-872)-2xFLAG or pRep1-*byr2*(65-180)-2xFLAG and cultured in MM+N without thiamine for 24 hours. Cells were induced for sexual differentiation by the plate mating assay system, and the levels of ppMAPK<sup>Spk1</sup>-GFP,

total MAPK^Spk1 and α-tubulin (loading control) were examined by Western blotting. Three biological replicates, shown in S13 Fig, were quantified using the Image Studio ver2.1 software (Licor Odyssey CLx Scanner). The data was analyzed by generalized linear mixed models and their least-squares means (± 1SE) are shown. Different letters (a, b and c) indicate significant differences (α = 0.05) among the three samples, pRep1-empty, pRep1-scd1-PB1 and pRep1-byr2-RBD, within each hour after induction of mating. When all three samples were deemed distinct, they were denoted differently: a, b and **c**. When all the samples were deemed indistinguishable, they were denoted with the same category: **a**. **(B)** Cell morphology and localisation of Cdc42^GTP, indicated by the CRIB-GFP signal, during the sexual differentiation process were compromised by the expression of Scd1-PB1 or Byr2-RBD. Cells harbouring *ras1.G17V* and CRIB-GFP (KT5938) were induced for mating/sexual differentiation by the plate mating assay condition (Materials and Methods), and live cell images of the CRIB-GFP signals were taken 16 hours after induction of the sexual differentiation. Navy arrows indicate cortical CRIB-GFP foci. Cells with the cortical CRIB-GFP foci (marked by an orange star) are counted as "cortical CRIB-GFP positive" cells. The scale bar is 10 μm. **(C)** Quantitation of the results presented in **(B)**. 300 cells were scored for cell deformity (left panel) and the cortical CRIB-GFP signal (right panel). % cells is presented. The experiment was repeated three times, and the mean and SD values of the % cells are plotted in the graphs. Ordinary one-way ANOVA followed by *post hoc* Dunnett's multiple comparisons test showed that the expression of scd1-PB1 or byr2-RBD reduces the % deformed cells and % cells with cortical CRIB-GFP. **(D)** GTP-loaded Ras1.G17V (1-172) directly binds to Byr2 (65-180) and Scd1 (760-872). *In vitro* GST pull-down assays of bacterially expressed Ras1.G17V (1-172), GST-Byr2 (65-180) and GST-Scd1 (760-872) were conducted as described in materials and methods. GTP-loaded Ras1.G17V (1-172) bound to both GST-Byr2 (65-180) and GST-Scd1 (760-872). **(E)** Two Ras1 effectors, Byr2 and Scd1, compete for GTP-loaded Ras1.G17V (1-172). *In vitro* GST pull-down assays of bacterially expressed Ras1.G17V (1-172) and GST-Byr2 (65-180) were conducted as in **(D)**. The addition of the Scd1 (760-872) fragment interfered with Ras1-Byr2 binding (the 4^th lane). Quantitated signal intensities of the Ras1.G17V (1-172) band in the gel are shown in the right panel.

Previous studies proposed that the adaptor protein Ste4 is involved in the activation of MAPKKK^Byr2 [33,39,40,63]. We therefore examined whether Ste4 is required for MAPK^Spk1 activation.

As was the case for the *byr2Δ* (S10F Fig), the loss of *ste4* abolished the expression of MAPK^Spk1 (S14C Fig), and we detected virtually no MAPK^Spk1 phosphorylation in the *ste4Δ* mutant (Figs 6A, yellow line and S14A), revealing that the adaptor^Ste4 is a prerequisite for the MAPK^Spk1 activation. The introduction of the *ras1.G17V* mutation neither restored the MAPK^Spk1 expression nor activation, nor mating (Figs 6A, 6B, and S14C). Thus, activation of Ras1 cannot take over Ste4 function. In a striking contrast, the *ste4Δ MAPKK^byr1.DD* double mutant induces the expression of MAPK^Spk1, constitutive MAPK^Spk1 activation and the "*paired*" phenotype as the *MAPKK^byr1.DD* single mutant cells (Figs 6A, 6B, and S14C). Collectively, a full MAPKKK^Byr2 activation requires both Ras1 and Ste4 functions.

Interestingly, Ste4's role is limited to the MAPKKK^Byr2 activation process as MAPKKK^Byr1.DD can suppress the *ste4Δ* mutant phenotype, whereas MAPKKK^Byr1.DD could not bypass the Ras1 function (Fig 3G).

### Ste6, a Ras1 GTP-GDP exchange factor, contributes to both the MAPK^Spk1 and the Cdc42 pathway activation

The activation of Ras1 is mediated by two GDP-GTP exchange factors (GEFs), Ste6 and Efc25 [64,65]. Ste6 is essential for mating but is dispensable during vegetative growth, whereas Efc25 is dispensable for mating but is required to maintain cell morphology during vegetative growth [64,65]. There has been an interesting proposition that Ste6 may specifically help Ras1 to activate the MAPK^Spk1 pathway but not the Cdc42 pathway, whilst Efc25 specifically facilitates Ras1 to activate the Cdc42 pathway [66]. We examined this hypothesis by monitoring the MAPK^Spk1 activation status and conducting genetic epistasis analysis of *ras1.G17V* and *MAPKK^byr1.DD* in the *ste6Δ* mutant.

In the *ste6Δ* cells, MAPK^Spk1 phosphorylation occurred at a detectable level (Figs 6C yellow line and S14B). When the *MAPKK^byr1.DD* mutation was introduced, the MAPK^Spk1 phosphorylation level was maximised (Figs 6C and S14B) and the nucleus MAPK^Spk1-GFP signal accumulated (Fig 6D). However, the sterile phenotype remained (Fig 6D). In contrast, the *ras1.G17V* mutation rescued the "pheromone-insensitive sterile" morphology of *ste6Δ*, as previously reported [64], exhibiting the "elongated" phenotype (Fig 6D) even though the increase of the ppMAPK^SPK1 accumulation was not as high as the case of the *MAPKK^byr1.DD* mutation (Figs 6C and S14B). The result indicates that, unlike the *ste4Δ* mutant, the mating deficiency of *ste6Δ* is not caused by a mere lack of MAPK^Spk1 activation but by a lack of Ras1 activation, which mediates *both* the MAPK^Spk1 and Cdc42 pathway activation in response to the pheromone signalling.

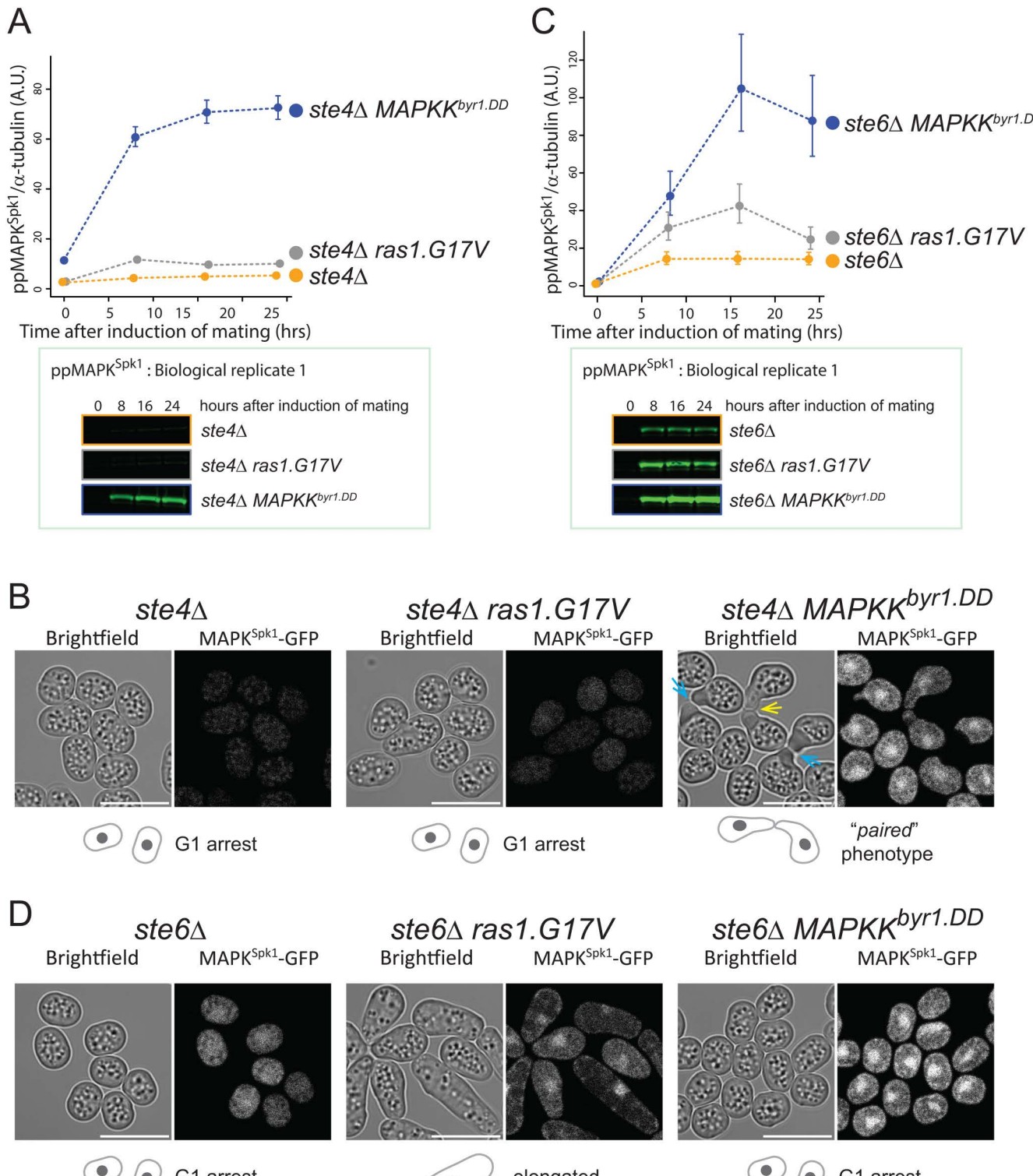

**Fig 6. Distinct contributions of Ste4 and Ste6 to MAPK$^{Spk1}$ activation. (A)** Ste4 is essential for MAPK$^{Spk1}$ activation. MAPK$^{Spk1}$ phosphorylation status in *ste4Δ* (KT4376), *ste4Δ ras1.G17V* (KT5143) and *ste4Δ MAPKKbyr1.DD* (KT5136) at times-points 0, 8, 16 and 24 hours after the induction of sexual differentiation were examined. Quantified results of three biological replicates (error bars are ±SEM) are presented. The ppMAPK$^{Spk1}$ Western blotting

result of Biological Replicate 1 is presented as a representative result where the ppMAPK$^{Spk1}$ levels are close to 0 in the *ste4Δ* (KT4376) and *ste4Δ ras1. G17V* (KT5143) strains. All the original Western blotting membranes are presented in S12A Fig. **(B)** The *MAPKK*byr1.DD but not *ras1.G17V* mutation suppresses the incapability of *ste4Δ* to cause pheromone-induced morphological change. Cell images of *ste4Δ* (KT4376), *ste4Δ ras1.G17V* (KT5143) and *ste4Δ MAPKK*byr1.DD (KT5136) strains were taken 24 hours after induction of mating. A yellow arrow indicates a typical "paired" phenotype example. Blue arrow indicates accumulated vacuoles at the conjugation tips as seen in the *MAPKK*byr1.DD mutant (Fig 1F). The scale bar represents 10 μm. **(C)** Lack of Ste6 does not result in the complete loss of MAPK$^{Spk1}$ phosphorylation. MAPK$^{Spk1}$ phosphorylation status in *ste6Δ* (KT4333), *ste6Δ ras1. G17V* (KT4998) and *ste6Δ MAPKK*byr1.DD (KT5139) at times-points 0, 8, 16 and 24 hours after the induction of sexual differentiation were examined. Quantified results of three biological replicates (error bars are ± SEM) are presented. The ppMAPK$^{Spk1}$ Western blotting result of Biological Replicate 1 is presented as a representative result where the ppMAPK$^{Spk1}$ level of the *ste6Δ* (KT4333) is clearly detectable. All the original Western blotting membranes are presented in S12B Fig. **(D)** The *MAPKK*byr1.DD but not *ras1.G17V* mutation suppresses the incapability of *ste6Δ* to cause pheromone-induced morphological change. Cell images of *ste6Δ* (KT4333), *ste6Δ ras1.G17V* (KT4998) and *ste6Δ MAPKK*byr1.DD (KT5139) strains were taken 24 hours after induction of mating. The scale bar represents 10 μm.

## Activation of Gpa1 represents the pheromone signalling

Gpa1, which plays the primary role in pheromone signalling [21], is expected to act upstream of Ste4 and Ste6. In agreement, in the *gpa1Δ* mutant, we detected no MAPK$^{Spk1}$ activation nor morphological change (Figs 7A, yellow line, 7B, left panel, and S15A). The *ras1.G17V* did not rescue the lack of *MAPK$^{Spk1}$* activation nor did it induce a shmoo-like morphological change (Figs 7A, grey line, 7B, the second right panel, and S15A), supporting our earlier observation that a Ras1-independent mechanism, involving Ste4, is essential for MAPK$^{Spk1}$ activation. Meanwhile, the *MAPKK$^{byr1.DD}$* caused the constitutive activation of MAPK$^{Spk1}$ (Figs 7A, blue line and S15A), but cells showed no morphological change (Fig 7B, the second left panel). When both *ras1.G17V* and *MAPKK$^{byr1.DD}$* mutations were introduced into the *gpa1Δ* strain, MAPK$^{Spk1}$ was activated, and a shmoo-like morphological change occurred (Figs 7A, red line, 7B, the right panel and S15A). Therefore, the activation of both the MAPK$^{Spk1}$ pathway and the Ras1 pathway is sufficient to take over the Gpa1 function.

The signalling hierarchy of Gpa1, Ras1 and the MAPK$^{Spk1}$ cascade was further validated in a heterothallic *h⁻* cell population. This population serves as a reference for measuring responses to nitrogen starvation without mating factor signalling as it lacks a mating partner and, consequently, the mating pheromone [41]. Interestingly, upon nitrogen starvation, the *h⁻* wildtype cells displayed a basal level of **pp**MAPK$^{Spk1}$ activation, although no morphological changes were observed (Figs 7C, yellow line, 7D, the left panel, and S15B). In contrast, *h⁻* cells with the constitutively active *gpa1.QL* mutation exhibited a "shmoo-like" morphological change as previously reported [21] and showed strong MAPK$^{Spk1}$ activation (Figs 7C, red line, 7D, the second left panel and S15B). Conversely, the *h⁻ ras1.G17V* mutant showed no apparent morphological alternation and exhibited only a basal level of **pp**MAPK$^{Spk1}$, comparable to that observed in the *h⁻* wildtype strain (Figs 7C, grey line, 7D, the middle panel, and S15B). Meanwhile, the *h⁻ MAPKK$^{byr1.DD}$* mutant induced strong constitutive MAPK-$^{Spk1}$ activation, confirming that the MAPKK$^{Byr1.DD}$ can activate MAPK$^{Spk1}$ regardless of the pheromone signal input (Figs 7C, blue line and S15B). Yet, the cell morphology remained unchanged (Fig 7D, the second right panel). Collectively, these results support the model in which Gpa1 serves as the central transducer of the pheromone signalling. It was noted that the Gpa1$^{QL}$-induced phenotype was largely dependent on Ras1 function as the *h⁻ gpa1.QL ras1Δ* double mutant exhibited only a basal level of **pp**MAPK$^{Spk1}$, and round cell morphology (Figs 7C, purple line, 7D, the right panel and S15B).

## Delayed negative feedback explains the transient increase of the total and phosphorylated MAPK$^{Spk1}$ in ordinary differential equations-based mathematical modelling

Our results, along with previous findings, can be summarised in a diagram illustrating that the pheromone signalling activates Ras1, leading to the activation of two effectors, MAPKKK$^{Byr2}$ and Cdc42-GEF$^{Scd1}$ (Fig 8A). In addition, the MAPK-KK$^{Byr2}$ activation also requires Ste4 function, and a reciprocal activation occurs between MAPK$^{Spk1}$ and Cdc42 for their full activation (Fig 8A).

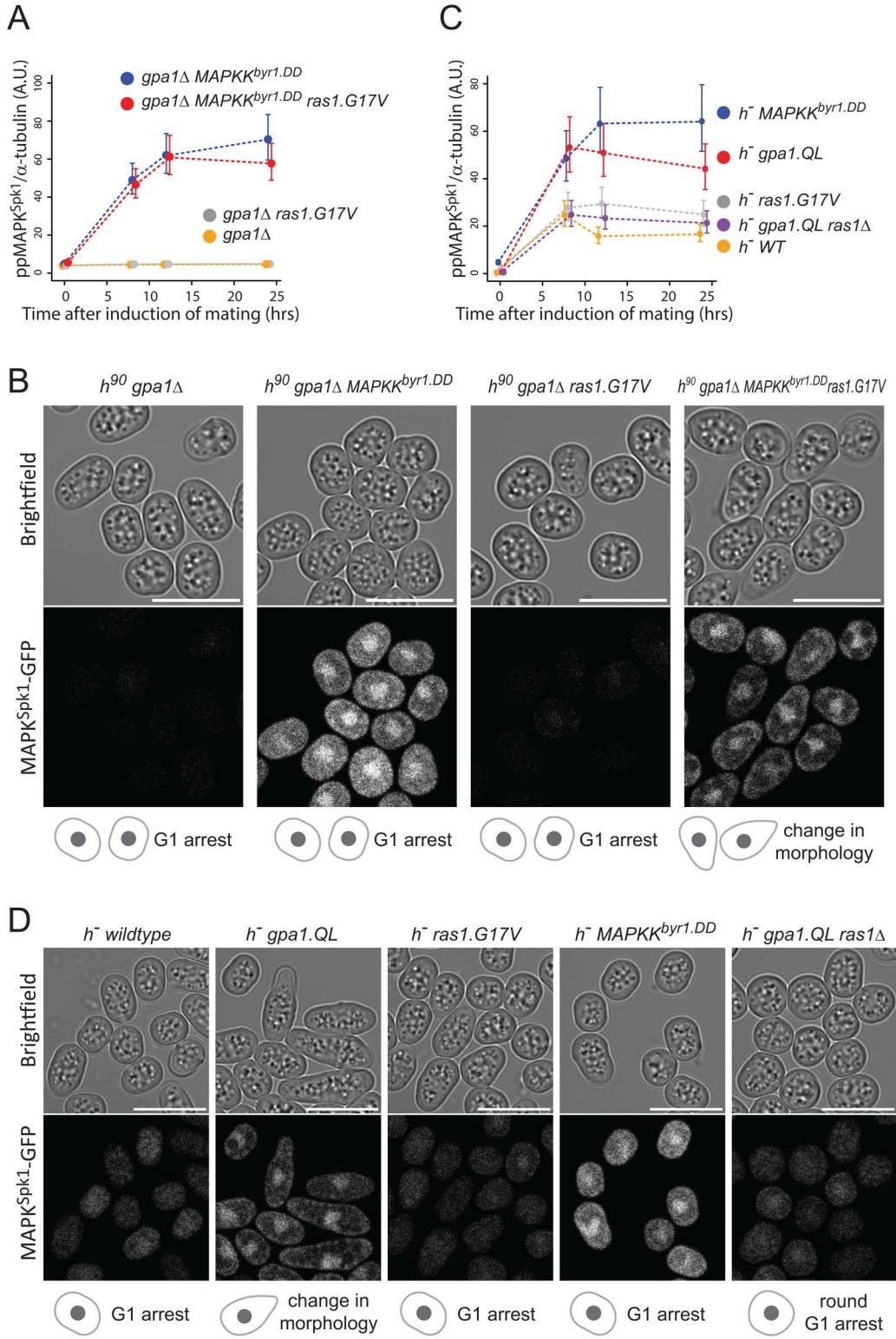

**Fig 7. Gpa1 plays the central role of the pheromone signal transduction by activating both MAPK<sup>Spk1</sup> and Ras1 pathways. (A)** MAPK<sup>Spk1</sup> phosphorylation status in homothallic *gpa1Δ* (KT4335), *gpa1Δ ras1.G17V* (KT5023), *gpa1Δ MAPKKbyr1.DD* (KT4353) and *gpa1Δ ras1.val17 MAPKKbyr1.DD* (KT5035) at times-points 0, 8, 12, 16 and 24 hours after the induction of sexual differentiation were examined. Results of three biological replicates

(error bars are ± SEM) are presented. Original Western blotting membranes are presented in S15A Fig. **(B)** Cell images of the above-mentioned strains at 16 hours after the induction of sexual differentiation. All the cell images were taken and processed as in Fig 1. The scale bar represents 10 µm. **(C)** MAPK$^{Spk1}$ phosphorylation status in *h- WT* (KT4190), *h- gpa1.QL* (KT5059), *h- ras1.G17V* (KT4233), *h- gpa1.QL ras1Δ* (KT5070) and *h⁻ MAPKK*byr1. DD (KT4194) at times-points 0, 8, 12 and 24 after the induction of sexual differentiation were examined. Results of three biological replicates (error bars are ± SEM) are presented. Original Western blotting membranes are presented in S15B Fig. **(D)** Cell images of the above strains at 12 h after the induction of sexual differentiation. All the cell images were taken and processed as in Fig 1. The scale bar represents 10 µm.

We next aimed to build a network structure that explains the experimental data we obtained. We are aware that whilst the wildtype data represents the entire sexual differentiation process, including G1 arrest, pheromone sensing, mating, followed by DNA recombination and sporulation, all the sterile mutants are blocked at the stage before successfully completing mating. Therefore, we need to be cautious when interpreting differences in the later timepoint data. However, we still decided to include all time points in our attempt to "describe" our experimental results using a single signalling framework. Towards this goal, we sought to determine whether the experimentally observed temporal profile of total MAPK$^{Spk1}$ ([tSpk1]), phosphorylated MAPK$^{Spk1}$ ([ppSpk1]) and active Cdc42 ([aCdc42]) (Figs 1C, E, G, 2A, 3B, 4 and S3)–in particular, the transient increase of ppSpk1 followed by attenuation–can be explained by specific network structures composed of the regulatory interactions identified and characterized in this study (see Material and Methods section and S16 and S17 Figs for details).

To address this question, we postulated up to 29 biochemical processes, each represented by a rate parameter *k*, to capture the documented relationships among the signalling components (S1 Table and Figs 8B, S16, S17A and S19A; see Material and Methods for details). Because many kinetic parameters of the pheromone signaling pathway have not been experimentally determined in fission yeast, model parameters were treated as effective quantities representing coarse-grained regulatory influences rather than elementary biochemical reaction rates, and the reported values therefore represent best-fit examples rather than unique or physiologically exact intracellular parameters. Initial parameter values were obtained using the MATLAB Optimization Toolbox (Mathworks, Natick, MA, USA) and were subsequently refined using a Markov chain Monte Carlo (MCMC) approach to improve agreement with the experimental data [67]. Importantly, the purpose of the modelling in this study is not to infer biologically exact kinetic parameter values, but to evaluate alternative candidate network structures based on their ability to account for this characteristic transient MAPK$^{Spk1}$ response, as well as the experimentally measured temporal dynamics of total Spk1 and active Cdc42.

Our initial model, Model A, did not yield a set of parameters that aligned with the experimental results (S17 Fig); [tSpk1] and [ppSpk1] did not decrease following an initial increase. To replicate the transient increase of [tSpk1] and [ppSpk1], a delayed negative feedback regulation was deemed necessary [68]. Given that the *MAPKK$^{byr1.DD}$* mutant entirely lacks downregulation (Fig 1E), we considered downregulations occurring downstream of MAPKK$^{byr1}$ (such as Pyp1 and Pmp1) to be physiologically insignificant. On the other hand, Sxa2 (a serine carboxypeptidase against a mating pheromone P-factor) and Rgs1 (a regulator of Gpa1), both of which are induced upon successful pheromone signalling [41,69–71], receptor internalization [72] and regulation of the *mapk$^{spk1}$* transcript or other components by antisense RNA [73] fit well to the criteria for the delayed negative feedback. We consolidated all these elements collectively as a single circuit and provided it with the rate constant $k_6$ (Model B, Fig 8B). Incorporating this negative feedback allowed us to obtain a set of parameters that effectively reproduced the transient increase of [tSpk1] and [ppSpk1] in wildtype, *ras1.G17V* and *scd1Δ* condition (Figs 8C and S18). Furthermore, the fit for the active Cdc42 was also significantly improved (Fig 8D).

## Additional regulations to further improve the model fit to the experimental results

Model B phenocopied the experimental results of tSpk1, **pp**Spk1 and active Cdc42 in wildtype and *scd1Δ* strains and qualitatively reproduced most data points in other strains. Notably, whilst Model A failed to reproduce the transient

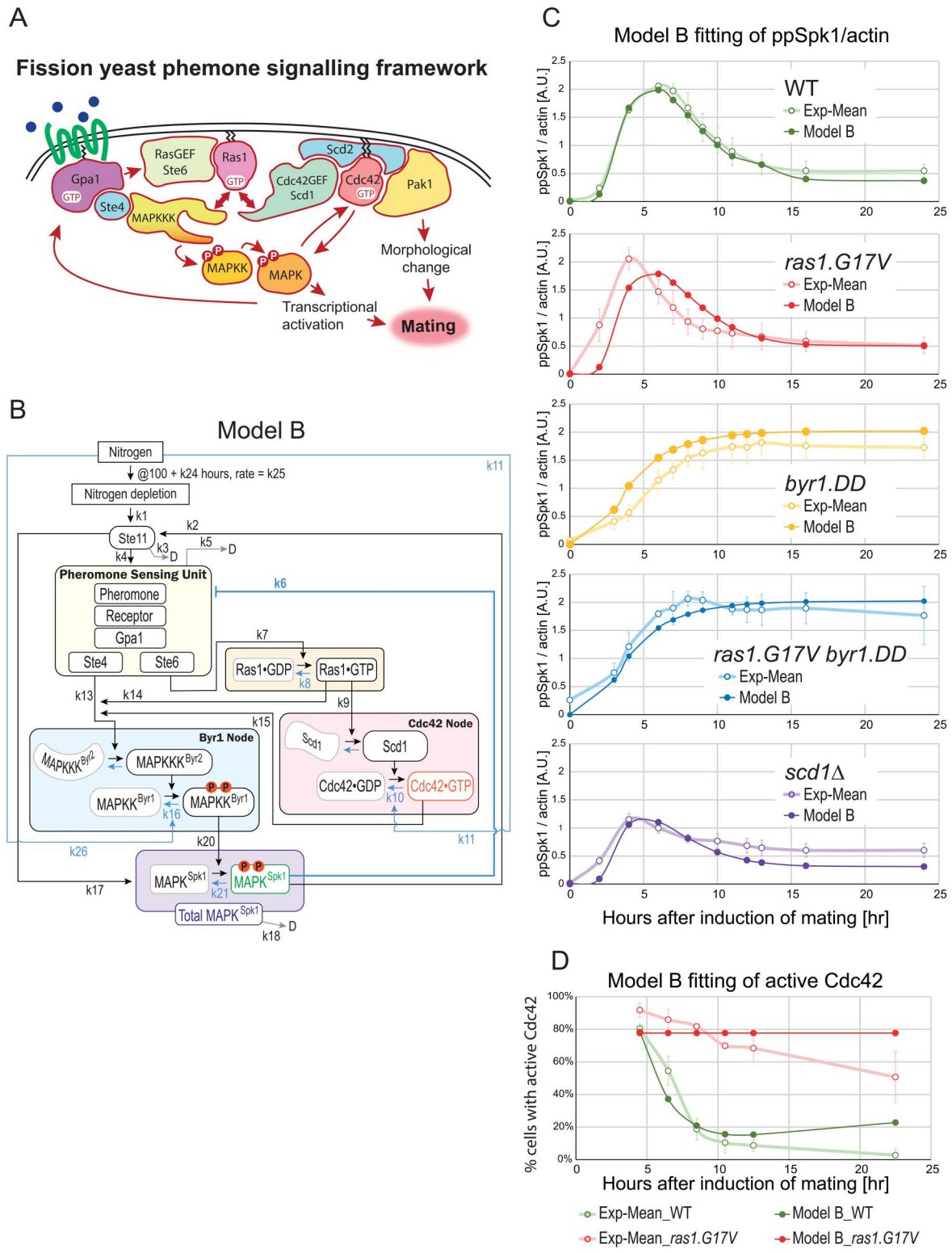

**Fig 8. Mathematical modelling of the fission yeast pheromone signalling dynamics. (A)** Schematic diagram of the fission yeast pheromone signalling pathway. The diagram highlights the bipartite mechanism to activate MAPKKK[Byr] [2], the branch for two Ras1 effectors, MAPKKK[Byr2] and Cdc-42GEF[Scd1], and the interplay between the MAPK[Spk1] and Cdc42 pathways. Successful mating requires both the MAPK[Spk1] and Cdc42 pathway activation.

**(B)** Model B components and framework. For the detailed implementation of the mutants, see S16 Fig and Materials and Methods. The measured components, total MAPK$^{Spk1}$, **pp**MAPK$^{Spk1}$ and Cdc42$^{GTP}$, are shown in purple, green and red, respectively. **(C)** Model B fittings for the **pp**MAPK$^{Spk1}$ levels in wildtype, *ras1.G17V*, *MAPKK*byr1.DD, *ras1.G17V MAPKK*byr1.DD and *Cdc42GEF*scd1Δ mutants. Mean values (open circles) and SD values (error bars) of the experimental results are shown in pale colours, and the model-fitted values are shown as filled circles in darker colours. Model B fittings for the total MAPK$^{Spk1}$ are shown in S18 Fig. **(D)** Model B fitting of the active Cdc42 levels in wildtype and *ras1.G17V* mutants. Mean values (open circles) and SD values (error bars) of the experimental results are shown in pale colours, and the model-fitted values are shown in filled circles in darker colours.

increase and subsequent attenuation of MAPK$^{Spk1}$ for any parameter set tested, Model B successfully reproduced these dynamics, indicating that the difference between the models is structural rather than parametric. However, Model B did not predict the early increase of the **pp**Spk1 levels in the *ras1.G17V* and *ras1.G17V MAPKK*$^{byr1.DD}$ strains (Fig 8C).

We revised Model B to generate Model C, where we introduced two aspects (S19 Fig). First, we added *ras1.G17V* strain-specific negative regulatory paths from **pp**Spk1 to modulate $k_9$ and $k_{14}$, the rate constants of two Ras1 downstream reactions. Second, to distinguish the outcomes between *MAPKK*$^{byr1.DD}$ and *MAPKK*$^{byr1.DD}$ *ras1.G17V* mutants, we added a positive regulation of **pp**Spk1 by Cdc42 (observed in the *Cdc42-GEF*$^{scd1}$Δ mutant, Fig 3B) with a rate constant $k_{27}$. As a result, model C reproduced both (i) the earlier **pp**Spk1 peak with the height comparable to the WT in the *ras1.G17V*, and (ii) the difference between *MAPKK*$^{byr1.DD}$ and *MAPKK*$^{byr1.DD}$ *ras1.G17V* strain (S19C Fig). The modelling analyses suggested that the constitutively active form of Ras1, Ras1.G17V, may have a qualitatively different biochemical activity compared to the wildtype Ras1.

## Prediction ability of our quantitative model

To examine the validity of Models B and C, we tested whether the models can predict the **pp**Spk1 levels of other experimental results, presented in Figs 3E, 3F, 6A, 6C, 7A and 7C, involving 20 strains of different genotypes (S20 Fig). Both Models B and C faithfully simulated Figs 3F and 6A, involving *MAPKK*$^{byr1.DD}$, *byr2Δ*, *ste4Δ* and *ras1.G17V*. Model B also produced a simulated outcome highly comparable to the results presented in Figs 3E and 7A, involving *MAPKK*$^{byr1.DD}$, *ras1Δ*, *gpa1Δ*, and *ras1.G17V*, whereas Model C's predictions for these experiments were less quantitative. On the other hand, Model C predicted Fig 7C results, involving heterothallic strains of wild type, *MAPKK*$^{byr1.DD}$, *gpa1.QL*, *ras1.G17V* and *ras1Δ*, better than Model B.

It was noted that both Model B and C predictions for Fig 6C experiment involving the strains with the *ste6Δ* mutation were the least accurate among all the tested experiments, although Model B correctly predicted high **pp**Spk1 for *ste6Δ MAPKKK*$^{byr1.DD}$ and low **pp**SPK1 for *ste6Δ* and *ste6Δ ras1.G17V* at a later stage of about 15 hours after induction of mating and Model C predicted a low **pp**Spk1 level in *ste6Δ*, and a high **pp**Spk1 level in *ste6Δ MAPKKK*$^{byr1.DD}$.

In summary, both models predicted the dynamics of **pp**Spk1 to a good extent in the 20 strains. However, Model C did not show a significantly improved prediction capability compared to Model B, despite two additional modelling parameters, $k_{28}$ and $k_{29}$. Therefore, the simpler Model B would represent the core framework of fission yeast pheromone signalling.

## The predicted negative feedback was experimentally validated in the *sxa2Δ* mutant cells

Our model predicts that a negative feedback circuit, impacting the pheromone production and/or sensing, plays a critical role in shaping the MAPK activation profile. One possible candidate to participate in such negative feedback is Sxa2, a carboxypeptidase that modulates the local concentration of one of the mating factors, P-factor [69,74]. We examined the effect of *sxa2* deletion in the *ras1.G17V* and *byr1.DD* mutant cells.

Strikingly, the ppMAPK$^{Spk1}$ level was substantially increased in the *sxa2Δ ras1.G17V* double mutant compared to the *ras1.G17V* single mutant (S21 Fig), validating the model prediction. As the MAPK$^{Spk1}$ activation profile of the *sxa2Δ ras1.G17V* double mutant still exhibits a peak (at 4 h) followed by a decline, rather than becoming constitutively active,

additional mechanism(s) still exist to fully compose the negative feedback circuit. Nonetheless, the result clearly demonstrates that Sxa2 plays a critical role, and modulating signal production is an effective way to shape MAPK[Spk1] signalling output even in the presence of the *ras1.G17V* mutation.

Intriguingly, the deletion of *sxa2* in the *byr1.DD* mutant did not substantially alter ppMAPK[Spk1] levels, but did induced the "elongated" morphology (S22 Fig). The "elongated" phenotype, characteristic of the *ras1.G17V* mutant that exhibits Cdc42 hyperactivation (Fig 4), might suggest that compromised negative feedback might have resulted in constitutively active Cdc42, as predicted for the wildtype case in Model A, which lacks the negative feedback (S17D Fig). These observations collectively support the existence of the predicted negative feedback mechanism.

## Discussion

By quantifying the MAPK[Spk1] and Cdc42 activation status during the mating process and conducting epistasis analysis between various signalling mutants, we established that Ras1 coordinates activation of two downstream pathways, the MAPK[Spk1] cascade and the Cdc42 pathway, and revealed that the *ras1.G17V* mutant phenotype (elongated shmoo formation) is caused by the prolonged activation of Cdc42, rather than constitutive activation of MAPK[Spk1]. We built a mathematical model to investigate the mechanism underlying the transient activation and subsequent attenuation of MAPK[Spk1]. The model incorporates a delayed negative feedback path acting on the MAPK[Spk1] axis, together with activation of the Byr1 Node by pheromone sensing unit (Ste4), Ras1 and the Cdc42 Nodes. The Cdc42-dependent activation of MAPK[Spk1] were also considered during model building. The model faithfully recapitulates MAPK[Spk1] and Cdc42 activation profiles in the wildtype and all mutant strains examined in this study. Furthermore, the predicted negative feedback impacting the pheromone production/sensing was experimentally validated.

The attenuated MAPK[Spk1] activation in the presence of Ras1.G17V indicated that an efficient feedback mechanism is in place to counteract the effect of Ras1.G17V, and we showed that Sxa2 plays a role in this regulation. The same trend has been reported in the mammalian models where oncogenic *KRAS* was expressed at the physiological levels [9–11]. Therefore, it is likely that the MAPK cascade attenuation is generally robust against upstream constitutive RAS signalling. As the MAPKK[byr1.DD] mutant caused constitutive activation of MAPK[Spk1], the effective negative regulation likely occurs upstream of, or at the same level as, MAPKK[Byr1]. In humans, ERK is shown to phosphorylate RAF proteins, the prototype MAPKKKs, to contribute to ERK signal attenuation [75–77]. Whether fission yeast MAPK[Spk1] can directly downregulate MAPKKK[Byr2] is an important future question.

We show that the adaptor[Ste4] is a prerequisite for MAPK[Spk1] pathway activation. Hence, the adaptor[Ste4] can be a target of the negative feedback loop against **pp**MAPK[Spk1]. This mechanism is shared by budding yeast, where Ste50, a Ste4 orthologue, modulates MAPKKK[Ste11] [78,79]. In humans, although an obvious Ste4/Ste50 orthologue is missing, multiple RAF-interacting proteins, including 14-3-3 proteins, as well as the formation of heterodimers between BRAF and CRAF, have been studied for their Ras-independent mechanism to activate RAF proteins [80]. Collectively, MAPK cascades retain a conserved feature of being resistant to oncogenic RAS mutations at physiological conditions.

Meanwhile, we found that MAPK[Spk1] activity is required for the Ras1.G17V-induced elongated shmoo formation (Fig 2C), and in turn, the Cdc42 function is also required for full activation of MAPK[Spk1] (Fig 3B), as previously proposed [33]. Therefore, the two Ras1 downstream pathways, Cdc42 and MAPK[Spk1], are not entirely separable. The situation is reminiscent of the *K-ras[G12D]* MEFs that show morphological anomalies without hyperactivation of ERK and AKT [9], where the MEFs still responded to inhibitors against MAPK and PI3K pathways, reverting the morphology to the one similar to wildtype [9]. It could be that a basal level of MAPK and PI3K pathway activation may be a prerequisite for the *K-ras[G12D]*–induced morphological anomalies.

The molecular mechanism of how MAPK[Spk1] contributes to Cdc42 activation will require further studies. As MAPK[Spk1] activates the master transcriptional regulator Ste11 [41,42,81], which up-regulates the pheromone signalling components, at least a part of the mechanism likely occurs through transcriptional regulation. In addition, the localisation of signalling

components may be regulated by MAPK[Spk1]. Interestingly, during vegetative growth, a stress-activated MAPK Sty1 negatively regulates Cdc42 even when protein synthesis was inhibited by cycloheximide, indicating direct phosphorylation events of Cdc42 and/or its regulators [82]. In budding yeast, MAPK[Fus3] brings Cdc24, the GEF for Cdc42, to the shmoo site by phosphorylating an adaptor protein Far1 [83], which otherwise sequesters Cdc24 into the nucleus [84,85]. Fission yeast does not have an obvious Far1 orthologue, but MAPK[Spk1] may also directly phosphorylate Cdc42 regulatory proteins such as Cdc42-GEF[Scd1], Scd2 or GAP-Cdc42[Rga4], all of which function at the shmoo site during the mating [36,59] (Fig 4A and 4B). In agreement with this hypothesis, a transient MAPK[Spk1] and MAPKK[Byr1] localisation on the cell cortex was observed during the mating process (Fig 1) [36]. Localisation of MAPK at the growing cell tips was also observed in other fungi, including *N. crassa* [86–89] and budding yeast, where MAPK[Fus3] can directly phosphorylate Bni1, a formin that organises actin filaments, to facilitate shmoo formation [90]. Furthermore, a recent phosphoproteome study using cancer cells carrying KRAS oncogenic mutations revealed ERK-dependent phosphorylation of Rho GTPase signalling components [91]. It will be important to fully uncover the role of MAPK at the plasma membrane in regulating cell morphology.

In this study, we also uncovered experimental evidence that two Ras1 effectors, MAPKKK[Byr2] and Cdc42-GEF[Scd1], can compete for Ras1. However, this observation was obtained under effector overexpression conditions, in which the abundance of effector molecules likely exceeded that of active Ras1. Future studies should examine the endogenous levels of active Ras1, Scd1 and Byr2 at the fusing cell tips to determine whether effector competition contributes to the regulation of the fission yeast RAS signalling. Such effector competition is likely to influence how Ras1 signalling is allocated between downstream pathways, thereby defining the physiological outcome. Although this competitive interaction was not explicitly addressed in the current mathematical model, extending the modelling framework to incorporate such competition will be an important future direction. Combined experimental and theoretical approaches may help clarify how Ras1 effector competition contributes to pathway robustness, signal prioritization, and cell fate decisions during mating.

In this study, we have revealed the vital contribution of Cdc42 to induce the *ras1.G17V* phenotype in fission yeast pheromone signalling. The prolonged Cdc42 activation is likely supported by the robust scaffold-mediated positive-feedback on Cdc42 [53,55]. In mammalian systems, small GTPases, such as Cdc42, RalA/B and Rac, play critical roles in oncogenic Ras signalling [13–18]. Therefore, oncogenic RAS-induced misregulation of small G proteins may be a common basis for the oncogenicity of mutated RAS-induced signalling. Targeting this process may, therefore, be an effective strategy against oncogenic RAS-driven tumourigenesis.

## Materials and methods

### Yeast strains and media

Genotypes of the *Schizosaccharomyces pombe* strains used are listed in the S2 Table. Gene disruption and 2xFLAG-GFP-tagging of genes were performed using the direct chromosomal integration method described previously [92–94]. Fission yeast media (YE, MM-N and SPA) and basic genetic manipulations are described in [95].

### Plate mating assay for synchronous mating

In order to induce synchronous mating, we established the "Plate mating assay" system as follows: Cells were grown in YE supplemented with adenine (0.2g/L) until the cell density reaches 8–10 x $10^6$ cells/ml. Cells were then washed with MM-N (1% glucose, supplemented with Leucine 40mg/L) using a filtration unit. These cells were resuspended to a cell density of 0.8-1x$10^7$cells/ml in MM-N. In the timecourse experiments, this moment was considered time 0. 8x$10^7$cells were re-filtered onto a PDVF membrane of 47mm diameter (Millipore DVPP04700) in order to evenly place the cells across the defined area. The filters bearing the cells over their surface were carefully transferred onto sporulation agar plates (SPA+Leucine 40mg/L, 50ml SPA agar/ 14cm diameter for 6 membranes), cell side up, and incubated at 30˚C. During the time course experiments, at each time-point, one membrane, which initially had 8x$10^7$cells at time 0, was removed from the SPA plate and placed into 5ml of ice-cold 20% trichloroacetic acid (TCA) in a 50ml falcon tube. The tube was shaken

vigorously to remove cells from the membrane. The tube was briefly centrifuged, and the membrane was removed from the tube. The tube was then centrifuged for 1 minute at 2000rpm to pellet cells. The supernatant was discarded, cells were resuspended in 1 ml 20% TCA and transferred to a tube capable of storage at -80˚C. The cells were pelleted again using a bench top centrifuge, supernatant discarded, and the pellet snap frozen in liquid nitrogen. Tubes were stored at -80˚C, ready for protein extraction. A cartoon representation of the outline of the plate mating assay procedure is presented in S1A Fig.

## Preparation of whole cell extracts

This method is based on that described by Keogh lab Protocols (https://sites.google.com/site/mckeogh2/protocols) with a modification to use Urea buffer at the final step to increase protein solubility. All steps were performed on ice unless otherwise stated. Cell pellets (typically $8\times10^7$ cells) were thawed on ice and resuspended in 250 µl 20% TCA. 250 µl of acid washed glass beads (SIGMA) were added and sampled chilled on ice for 5 minutes. Cells were broken in a FastPrep24 cell beater (MP Biomedicals, speed 6.5 x 1 minute x 3 with 1 minute intervals). Cell extracts were collected by piercing the bottom of tubes with a needle and placing them into collection tubes and centrifuged at 2000rpm for 1 minute. Beads were washed with 300 µl of 5% TCA and centrifuged at 2000rpm for 1 minute, this was repeated twice. The contents of the collection tube were transferred to a 1.5ml tube and centrifuged at 14K rpm for 10 minutes at 4°C. The liquid was discarded and the cell pellet washed with 750 µl of 100% ethanol. Tubes were centrifuged briefly (~1min 14000rpm) again. Ethanol was completely removed with a pipette. The pellet was then resuspended in 100 µl of Urea buffer (8M urea, 500mM Tris pH9.4, 1% SDS, 5mM EDTA). The sample was centrifuged and 40 ul of the supernatant was mixed with 80 µl of 3x protein loading buffer [96]. The sample was then heated for 5 minutes at 95˚C and stored at -80˚C until use.

## Western blotting

Protein extracts were subject to SDS-PAGE and were transferred to Immobilon-FL PVDF membrane (Millipore). Membranes were blocked for a minimum of 1-hour in Odyssey Blocking Buffer (OBB) (Li-cor) diluted 1:1 in PBS. Primary antibody incubation was carried out overnight in OBB 1:1 in PBS with anti phospho-p44/42 MAPK (Erk1/2) (Thr202/Tyr204) rabbit monoclonal antibody (#4370 Cell Signaling Technology. Used at 1:4000 dilution), anti GFP antibody ((0.4mg/ml), Roche Cat No 11814460001. Used at a 1:2000 dilution), anti FLAG M2 antibody (Sigma F1804, 1µg/µl, used at 1:2000 dilution), anti actin antibody (Life Technologies MA1–744; mouse monocolonal RRID:AB_2223496, 1:2000 dilution) and anti α-tubulin antibody, TAT1 (generous gift from Keith Gull, 1:3000). Membranes were washed 3x10 minutes with 20ml TBST (50mM Tris, pH 7.4 – 150mM NaCl – 0.1% Tween 20) whilst gently shaking followed by secondary antibody incubation (IRDye 680LT goat anti-mouse antibody, Li-cor 926–32211(1.0mg/ml), 1:16,000 dilution and IRDye 800CW goat anti-rabbit secondary antibody Li-cor 926–68020 (1.0mg/ml), 1:16000 dilution in 0.01% SDS, 0.1% Tween 20 in OBB 1:1 PBS) for 1 hour in the dark to prevent fluorophore bleaching. This was followed by 2x 10 minute TBST washes and a 1x 10 minute TBS (50mM Tris, pH 7.4 – 150mM NaCl) wash before scanning on the Odyssey CLx infrared Imaging System (Li-cor). Protein quantitation was conducted using either the Image Studio V2.1 software supplied with the Odyssey CLx scanner or Fiji (Image J, [97]).

## Quantitation of MAPK^Spk1 phosphorylation

To detect the activated MAPK^Spk1, whole cell extracts were prepared and Western blotting analysis was carried out using the anti-phospho ERK antibody (#4370 Cell Signaling Technology) that recognises the dual phosphorylation at the conserved TEY motif. Specificity of the antibody against the phosphorylated MAPK^Spk1 (**pp**MAPK^Spk1) in the fission yeast cell extracts was confirmed before proceeding with further experiments as shown in S1 Fig. As described above in the "**Plate mating assay for synchronous mating**", during the time course experiments, at each time-point, one membrane, which initially had $8\times10^7$ cells at time 0, was used to prepare the whole cell extracts. Then an equal volume of each sample was

loaded to the Western blotting. In this manner, we observed that the change of the protein level throughout the mating process per a defined starting material. Recent proteiomis studies showed that during the sexual differentiation, both actin and α-tubulin levels stay constant until sporulation starts [98]. During the sporulation process, actin stays constant whereas the level of α-tubulin decreases [98]. Therefore, we used actin as an interanal control when the Western blotting involves wildtype cells, which undergo sporulation. We also used α-tubulin as an internal control when sporulation deficient mutant strains were compared each other. The benefit of α-tubulin was that the signal intensity was stronger and therefore, quantitation can be more sensitive. For all Western blotting analysis, three biological replicates of time-course experiments were used.

## Cell imaging

A Zeiss LSM 980 Airyscan 2 microscope equipped with Plan-Apochromat 63x/1.40 Oil DIC f/ELYRA objective in LSM confocal mode was used for cell imaging to capture the morphology of cells and the localisation of the GFP-tagged MAPK[Spk1] during mating time-courses. Each time point comprises 32 serial images with 0.24 μm intervals along the Z axis taken to span the full thickness of the cell. Images were cropped and combined using Fiji [97].

For a higher resolution images used to quantify the GFP-tagged MAPK[Spk1] singals (presented in S4 Fig), the Zeiss LSM 980 Airyscan 2 microscope was used in Airyscan multiplex-4Y mode. Each image comprises 47 serial images with 0.15 μm intervals along the Z axis taken to span the full thickness of the cell.

For cell images presented in Figs 4A, 5, S5 and S8, a 2D array scanning laser confocal microscope (Infinity 3, VisiTech) on a NikonTi-E microscope stand equipped with a Hamamatsu Flash 4.0V2 sCMOS camera and a Plan Apo 100x/1.45 objective was used. Each time point comprises 24 serial images with 0.25 μm intervals along the Z axis taken to span the full thickness of the cell. All of the Spk1-GFP images were deconvolved using the Huygens Essential Deconvolution software (Scientific Volume Imaging). Deconvolved images were cropped and combined using Fiji [97].

## Quantification of GFP-tagged MAPK[Spk1] signals

In order to quantify the MAPK[Spk1]-GFP signals throughout the sexual differentiation stages of wildtype cells, cells carrying the MAPKSpk1-GFP and Smd2-tdTomato were imaged using the Zeiss LSM 980 Airyscan 2 microscope in Airyscan multiplex-4Y mode. These cells were induced for sexual differentiation for at least 8 hours, so that the GFP signals were clearly detectable and could be used to define the total area, comprising of both cytoplasm and nucleus. To define the area of the nucleus, tdTomato-tagged Smd2 (small nuclear ribonucleoprotein) was used. Using these two signals, 3D image segmentation was conducted using Imaris (Oxford Instruments) to generate two segments, Total and Nucleus. We then measured the MAPK[Spk1]-GFP signal intensity in each segment. The process was summarised in S4B Fig.

## Quantification of CRIB-GFP signal

CRIB-GFP signal was used to examine activation status of Cdc42. For images presented in Fig 4A, 25 serial Z images with step size of 0.25 μm were taken. Images were deconvolved using Huygens Essential (Scientific Volume Imaging) and maximum intensity projections are presented. Deconvolved images were Z-projected (maximum intensity), cropped and combined using Fiji [97].

For images presented in S9A Fig, for each image, 30–45 serial images with 0.2 μm interval along the Z axis were taken to span the full thickness of the cell. Using FIJI software maximum intensity Z-projections were created and used for quantitation. Intensity of GFP signal on the cell cortex was measured using FIJI, by applying line measurement with a line width of 1 μm which traced the inner cell cortex. The measurement was done along one of the cell tips that shows a stronger GFP signal as indicated in an example image in S9B Fig. 40 cells without septum were measured for each strain. Quantified CRIB-GFP intensity traces were analysed in R (R Core Team http://www.R-project.org/). First, rolling averages over +/-5 measurement points were calculated. Next, the X-coordinates of these traces were aligned by their peak intensity.

Finally, the average curve from all aligned traces per strain was calculated, and displayed in S9B Fig with respective standard error of the mean curves (dashed lines).

## Glutathione-S-transferase (GST) tag pull-down assays

Fission yeast cDNA fragments encoding Ras1.G17V (1–172), Byr2 (65–180) and Scd1 (760–872) were cloned in the pLE-ICS1 (Ras1.G17V) and pLEICS4 (Byr2 and Scd1) vectors (Protex [Protein Expression Laboratory], University of Leicester) to produce N-terminally His6-tagged Ras1.G17V (1–172), N-terminally GST-tagged Byr2 (65–180) and N-terminally GST-tagged Scd1 (760–872). Constructs were expressed in *E.coli* BL21 Rosetta cells in LB. All bacteria cell lysates were prepared in phosphate-buffered saline (PBS), pH 7.4.

His6-Ras1.G17V (1–172) was purified using Ni Sepharose 6 Fast Flow (GE Healthcare) and dialysed into buffer A (20 mM Tris-Cl (pH 7.5), 100 mM NaCl, 5mM MgCl2, 1mM β-mercaptoethanol). For GTP loading, buffer A of the Ras1.G17V (1–172) preparation was firstly replaced by Exchange buffer (20 mM Tris-Cl (pH 7.5), 1mM EDTA, 1mM β-mercaptoethanol) using a filtration unit (Amicon Ultra Centrifugal Filter Unit, 10K cut off, Millipore). GDP-GTP exchange reaction was carried out by adding GTP and EDTA to a final concentration of 8 mM and 12 mM respectively and incubating the sample at 37°C for 10 min. The exchange reaction was terminated by adding MgCl2 to a final concentration of 20.5 mM. The resultant GTP-loaded Ras1.G17V (1–172) was washed with buffer A using the filtration unit and immediately used for the GST pull-down assays.

GST-Byr2 (65–180) and GST-Scd1 (760–872) were purified using glutathione sepharose 4B (GE Healthcare). The proteins bound on the glutathione beads were washed with buffer A and used for the GST pull-down assays together with the Ras1.G17V (1–172) preparation. For the competition assays shown in Fig 5E, Scd1 (760–872) fragment was prepared by cleaving it out from the N-terminal GST, which was bound to the glutathione beads, using TEV protease.

GST pull-down assays were carried out by incubating Ras1.G17V (1–172) (GDP bound or GTP bound) and GST-Byr2 (65–180) or GST-Scd1 (760–872) with or without Scd1 (760–872) fragment at 4 °C for 30 min. Supernatant of the reaction was saved and mixed with the same volume of the 3x loading buffer [96] to generate the "unbound" fraction. The remaining beads were washed twice, resuspended in buffer A of the original volume and mixed with the same volume of the 3x loading buffer to generate the "bound" fraction. Protein samples were run on a SDS-PAGE gel and proteins were visualised by InstantBlue protein stain (Expedeon). Quantitation of the intensities of Ras1.G17V (1–172) bands in Fig 5E was carried out using Fiji software.

## Statistical analysis

Western blotting data presented in S2 Fig was quantified as stated in the "Western blotting" section, and individual replicate results are presented in S3A, S3B and S3C Fig. The mean (± SD) of the ppSpk1/actin triplicate data is presented in Figs 1C, 1E, 1G, 2A, and 3B. The data was analyzed by the generalized additive mixed model (GAMM) by fitting GAMMs on temporal variations in the data points (ppSpk1/actin, total Spk1/actin, and ppSpk1/total Spk1) for individual strain using the mgcv package in R. We assumed that the data were distributed according to a normal distribution (identity link). To account for the repeated measurements of data, the models included the genotype of each strain as a random factor. The smoothed curves were back-transformed from the GAMMs and drawn in the figures with twofold standard errors (S3D Fig).

Western blotting data presented in S13 Fig, which was quantified and presented in Fig 5A, was analyzed by the generalized linear mixed models (GLMMs), which involved gamma distribution (ln-link). The models included three samples, "pRep1-empty", "pRep1-scd1-PB1", "pRep1-byr2-RBD", and "hours after induction of mating" as fully crossed fixed factors. Mixed models were necessary for these models to account for the repeated measurements from each Western blot membrane data.

To analyse the data presented in S4 Fig, Prism (GraphPad) was used.

## Mathematical modelling

We constructed models with a system of 32 ordinary differential equations (ODEs) describing the amount of 7 molecular components (Ste11, Ste4/6, active Ras1 ([aRas1]), active Cdc42 ([aCdc42]), phosphorylated active MAPKK$^{Byr1}$ ([pByr1]), total MAPK$^{Spk1}$ ([tSpk1]) and phosphorylated **pp**MAPK$^{Spk1}$ ([ppSpk1])), the nutrient status ([N]) and time (t) in 5 different genetic conditions (wildtype, *ras1.G17V*, *MAPKK$^{byr1.DD}$*, *ras1.G17V MAPKK$^{byr1.DD}$* and *scd1Δ*) (S16 Fig). All the signalling components were set to be in close proximity, based on our own observation (Figs 1-4) and a series of localisation studies [35,36,38].

We postulated up to 29 biochemical processes involving the signalling components (S1 Table and Figs 8B, S17A and S19A) to reflect reported relationships among the signalling components. Each of these processes is accompanied by a rate parameter *k*. We first obtained an initial guess of the parameter values using an optimization toolbox of MATLAB software (Mathworks, Natick, MA, USA) and fitted the parameter values to better reproduce the experimental results using a Markov chain Monte Carlo method (MCMC) [67].

***Modelling of regulatory networks for Spk1:***

**The initial model (Model A).** Model A (S17A Fig) consists of the following differential equations for the control (wild-type):

$$[N] = \frac{1}{\pi}\left[arctan\left\{\frac{t-(100+k_{24})}{k_{25}}\right\} + \frac{\pi}{2}\right]$$

(1)

Equation (1) describes the temporal change in nutrient availability. The variable [N] represents the nutrient (nitrogen) status of the medium and is modeled as a smooth step-like function of time. Initially, [N] remains close to zero, corresponding to nutrient-rich conditions. After a characteristic time point, [N] rapidly increases toward one, representing nitrogen depletion. The arctangent function is used to approximate a stepwise transition while avoiding a discontinuous jump. The parameter $100+k_{24}$ determines the timing of nitrogen depletion, and $k_{25}$ controls the steepness of the transition. This formulation allows us to model nitrogen starvation as a gradual but rapid change in nutrient status, consistent with experimental conditions.

$$\frac{d[Ste11]}{dt} = k_1[N] + k_2[ppSpk1]/k_{12} - k_3[Ste11]$$

(2)

Equation (2) describes the dynamics of Ste11, a master transcriptional regulator of the pheromone signalling pathway. The production of Ste11 is induced by nitrogen starvation, represented by the nutrient-dependent term $k_1[N]$, consistent with previous reports showing activation of Ste11 under nitrogen-depleted conditions [99,100]. In addition, Ste11 is upregulated by phosphorylated MAPK$^{Spk1}$ (ppSpk1) [42] through a positive feedback loop, represented by the term $k_2[ppSpk1]/k_{12}$. Here, $k_{12}$ is a normalization constant used to relate the model variable [ppSpk1] to the experimentally measured Western blot signal; a detailed explanation is provided below. Ste11 is degraded or inactivated at a constant rate proportional to its abundance ($k_3[Ste11]$). Through this formulation, Ste11 integrates nutrient status and MAPK signalling activity to control the expression of downstream pheromone signalling components.

$$\frac{d[Ste4/6]}{dt} = k_4[Ste11] - k_5[Ste4/6]$$

(3)

Equation (3) describes the dynamics of the pheromone sensing unit (PSU). The PSU comprises genes encoding pheromones, receptors, Gpa1, Ste4, and Ste6 that are transcriptionally regulated by Ste11 and collectively represent the expression and availability of core pheromone signalling components [41,100–103]. The production of the PSU is driven

by Ste11, the master transcriptional regulator, and is modeled by the term $k_4$[Ste11]. The PSU is removed or inactivated at a constant rate proportional to its abundance ($k_5$[Ste4/6]), representing turnover and dilution of the signalling components. This formulation reflects the transcriptional control of pheromone signalling components by Ste11 and their constitutive turnover.

$$\frac{d[aRas1]}{dt} = k_7[Ste4/6]\left(1 - [aRas1]\right) - k_8[aRas1]$$

(4)

Equation (4) describes the activation dynamics of Ras1. Active Ras1 ([aRas1]) is produced in response to the phero-mone sensing unit (Ste4/6), reflecting Ste6-dependent activation of Ras1 following pheromone signalling [64]. This activation is modeled by the term $k_7$[Ste4/6](1-[aRas1]), where the factor (1-[aRas1]) ensures saturation as Ras1 approaches its maximal activation level. Active Ras1 is inactivated at a constant rate proportional to its abundance ($k_8$[aRas1]), representing GTP hydrolysis and other inactivation processes. This formulation captures stimulus-dependent Ras1 activation with a finite maximal level and constitutive inactivation. In this model, active Ras1 serves as a shared upstream activator of both Cdc42 and Byr1 Nodes, without explicitly modelling competitive binding or exclusivity between the two effectors.

$$\frac{d[aCdc42]}{dt} = k_{19}\left\{k_9[aRas1]\left(1 - \frac{[aCdc42]}{k_{19}}\right) - k_{10}\left\{1 + k_{11}\left(1 - [N]\right)\right\}\frac{[aCdc42]}{k_{19}}\right\}$$

(5)

Equation (5) describes the dynamics of active Cdc42 ([aCdc42]). Activation of Cdc42 is driven by active Ras1 through the guanine nucleotide exchange factor Scd1 [32], and is modeled by the term $k_9$[aRas1] (1 - [aCdc42]/$k_{19}$). Here, $k_{19}$ is a normalization constant that relates the model variable [aCdc42] to the experimentally measured CRIB-GFP signal; a detailed explanation is provided below. The saturation factor ensures that Cdc42 activity does not exceed its maximal level. Active Cdc42 is negatively regulated by constitutive inactivation processes, represented by the term $k_{10}$[aCdc42]/$k_{19}$. In addition, Cdc42 is subject to enhanced inactivation during vegetative growth in the presence of nitrogen [104], reflecting higher activity of GTPase-activating proteins under nutrient-rich conditions; this regulation is captured by the nitrogen-dependent term $k_{11}$(1 - [N]) [aCdc42]/$k_{19}$. Together, this formulation models Ras1-dependent activation of Cdc42 balanced by nutrient-dependent and nutrient-independent inactivation processes.

$$\frac{d[ppByr1]}{dt} = k_{13}[Ste4/6]\left(1 + k_{14}[aRas1] + k_{15}\frac{[aCdc42]}{k_{19}}\right)\left(1 - [ppByr1]\right) - k_{16}\left\{1 + k_6\left(1 - [N]\right)\right\}[ppByr1]$$

(6)

Equation (6) describes the dynamics of the Byr1 Node, which represents the combined activity of MAPKKK[Byr2] and MAPKK[Byr1] in the MAPK signalling cascade. Activation of the Byr1 Node is driven by the pheromone sensing unit (Ste4/6), reflecting Ste4-dependent activation of MAPKKK[Byr2] [33,39,40,63], and is modeled by the term $k_{13}$[Ste4/6](1-[ppByr1]). In addition, the activity of the Byr1 Node is positively modulated by active Ras1 and active Cdc42 [33], represented by the terms $k_{14}$[aRas1] and $k_{15}$[aCdc42]/$k_{19}$, respectively. These terms summarize experimentally reported regulatory influences of Ras1 and Cdc42 on MAPK pathway activation and represent effective modulation rather than direct elementary bio-chemical reactions. The Byr1 Node is subject to constitutive inactivation or turnover, represented by the term $k_{16}$[ppByr1]. Furthermore, MAPK signalling is negatively regulated under nutrient-rich conditions [105]; this effect is incorporated as nitrogen-dependent inhibition of the Byr1 Node, represented by the term $k$(1-[N])[ppByr1]. Together, this equation models integration of pheromone-dependent activation, small GTPase–mediated modulation, and nutrient-dependent repression to control MAPK pathway activity.

$$\frac{d[tSpk1]}{dt} = k_{17}[Ste11] - k_{18}[tSpk1]$$

(7)

Equation (7) describes the dynamics of total MAPK$^{Spk1}$ ([tSpk1]). The production of Spk1 is induced by Ste11, reflecting transcriptional upregulation of MAPK components during the mating response, and is modeled by the term $k_{17}$[Ste11]. Total Spk1 is removed at a constant rate proportional to its abundance ($k_{18}$[tSpk1]), representing protein turnover and dilution. This formulation captures Ste11-dependent expression and constitutive turnover of MAPK$^{Spk1}$.

$$\frac{d[ppSpk1]}{dt} = \left(k_{20}[ppByr1]\right)\left([tSpk1] - \frac{[ppSpk1]}{k_{12}}\right) - k_{21}\frac{[ppSpk1]}{k_{12}}$$

(8)

Equation (8) describes the dynamics of phosphorylated MAPK$^{Spk1}$ ([ppSpk1]). Phosphorylation of Spk1 is driven by the activity of the Byr1 Node, which represents the combined MAPKKK/MAPKK activity in the MAPK cascade, and depends on the availability of unphosphorylated Spk1. This process is modeled by the term $k_{20}$[ppByr1]([tSpk1]-[ppSpk1]/$k_{12}$). Phosphorylated Spk1 is removed through dephosphorylation and other inactivation processes, represented by the term $k_{21}$[ppSpk1]/$k_{12}$. Together, this equation models MAPK activation as a Byr1-dependent phosphorylation reaction balanced by constitutive inactivation.

Note that [tSpk1] represents the amount of total Spk1, as measured by the signal intensity of the total Spk1 Western blot. Accordingly, the amounts of other model variables representing protein abundance (e.g., [Ste11]) are expressed as relative values normalized to 1 unit of [tSpk1], except for [ppSpk1] and [aCdc42]. In the case of phosphorylated Spk1, [ppSpk1] represents the signal intensity of phospho-specific Spk1 Western blot (and not that of total Spk1). The amount of phosphorylated Spk1 normalized to 1 unit of total Spk1 is therefore given by [ppSpk1]/$k_{12}$. In case of active Cdc42, [aCdc42] represents the experimentally measured CRIB-GFP signal (Fig 4B). The amount of active Cdc42 normalized to 1 unit of total Spk1 is given by [aCdc42]/$k_{19}$.

For *ras1.G17V* mutant, we used the same differential equations except that [aRas1] was fixed to a constant value of $k_{22}$. Similarly, for *byr1.DD* mutant, [ppByr1] was fixed to a constant value of $k_{23}$. For scd1-delta mutant, [aCdc42] was fixed to zero.

**Model B.** In Model B (Fig 8B), a negative regulation from ppSpk1 to Ste4/6 was added to Model A. Eq. (3') was used instead of Eq. (3) in Model A.

$$\frac{d[Ste4/6]}{dt} = k_4[Ste11] - k_5\left(1 + k_6[ppSpk1]/k_{12}\right)[Ste4/6]$$

(3')

In equation (3'), the additional term $k_6$ [ppSpk1]/$k_{12}$ reduces the effective level of Ste4/6 in proportion to MAPK activity. This term represents delayed negative feedback from MAPK signalling to upstream pheromone sensing components, summarizing MAPK-dependent attenuation mechanisms acting on the PSU.

**Model C.** Model C (S19A Fig) extends Model B by introducing additional regulatory interactions that are specific to the *ras1.G17V* background. In this model, two negative regulations from ppSpk1 to downstream reactions of Ras1 are incorporated to account for experimentally observed attenuation of signalling in the presence of constitutively active Ras1. These regulations are implemented by replacing Eqs. (5) and (6) with Eqs. (5') and (6'), respectively, only under *ras1. G17V* and *ras1.G17V byr1.DD* conditions. In these conditions, [aRas1] and [ppByr1] are fixed to constant values ($k_{22}$ and $k_{23}$, respectively), reflecting constitutive Ras1 activation and Byr1 activation in the corresponding mutants.

$$\frac{d[aCdc42]}{dt} = k_{19}\left\{k_9[aRas1]\left(1 - \frac{[aCdc42]}{k_{19}}\right)/\left(1 + k_{28}[ppSpk1]\right) - k_{10}\left\{1 + k_{11}\left(1 - [N]\right)\right\}\frac{[aCdc42]}{k_{19}}\right\}$$

(5')

Equation (5') modifies Eq. (5) by introducing a MAPK$^{Spk1}$-dependent negative regulation on Cdc42 activation that is specific to the *ras1.G17V* background. The additional term $k_{28}$ [ppSpk1] represents attenuation of Ras1–Scd1–Cdc42 signalling by MAPK activity, thereby reducing Cdc42 activation in proportion to MAPK$^{Spk1}$ phosphorylation. This modification

captures MAPK-dependent feedback acting on the Ras1–Cdc42 branch under constitutive Ras1 activation, while the remaining terms are identical to those in Eq. (5).

$$\frac{d[ppByr1]}{dt} = k_{13}[Ste4/6]\left\{1 + k_{14}[aRas1]/\left(1 + k_{29}[ppSpk1]\right) + k_{15}\frac{[aCdc42]}{k_{19}}\right\}$$
$$\left(1 - [ppByr1]\right) - k_{16}\left\{1 + k_{26}\left(1 - [N]\right)\right\}[ppByr1]$$

(6')

Equation (6') modifies Eq. (6) by adding a MAPK$^{Spk1}$-dependent negative regulation on the Ras1-mediated activation of the Byr1 Node. The additional term $k_{29}[ppSpk1]$ represents attenuation of Ras1–Byr2 signalling by MAPK activity, which is applied specifically under *ras1.G17V* and *ras1.G17V byr1.DD* conditions. This modification implements MAPK-dependent suppression of Ras1-driven MAPK pathway activation under constitutive Ras1 signalling, while all other terms remain unchanged from Eq. (6).

In addition, a regulation to activate aByr1 to ppSpk1 by aCdc42 was added to expect the distinct outcome of *byr1.DD* and *ras1.G17V byr1.DD* as observed in the experiment. Eq. (8') was used instead of Eq. (8).

$$\frac{d[ppSpk1]}{dt} = \left(k_{20} + k_{27}\frac{[aCdc42]}{k_{19}}\right)[ppByr1]\left([tSpk1] - \frac{[ppSpk1]}{k_{12}}\right) - k_{21}\frac{[ppSpk1]}{k_{12}}$$

(8')

Equation (8') modifies Eq. (8) by adding a Cdc42-dependent enhancement of MAPK$^{Spk1}$ phosphorylation, represented by the term $k_{27}[aCdc42]/k_{19}$. Apart from this additional term, the formulation of MAPKSpk1 phosphorylation and dephosphorylation remains unchanged from Eq. (8).

**Initial guess of the parameter values.** We first obtained an initial guess of the parameters by using an optimization toolbox of MATLAB software (Mathworks, Natick, MA, USA). We found parameters that optimize an ordinary differential equation (ODE) in the least-squares sense, using the problem-based approach, according to the software website (https://jp.mathworks.com/help/optim/ug/fit-ode-problem-based-least-squares.html?lang=en) within the range of $0.1 \leqq k_i \leqq 10$, where $k_i$ represents the *i*-th fitting parameter. The parameters obtained in this way did not reproduce the transient increases of [tSpk1] and [ppSpk1] as observed in our experiment, and thus used as the prior for the MCMC approach. The codes are available on our Github (https://github.com/akkimura/pombePheromoneSignalling2025_250423f).

**Fitting with an MCMC method.** We used a Metropolis algorithm for a Markov chain Monte Carlo (MCMC) method [67] to search parameters reproducing the experimental behaviour using the values of initial guess obtained with MATLAB as the prior. In the algorithm, we change each parameter values by multiplying the previous value (or initial guess for the first step) with exp($r$), where $r$ is a random number following the normal distribution with the mean of 0 and standard deviation of 0.01. We adopted the latter value of 0.01 by testing different values and checked the efficiency of the fitting.

The likelihood of the fitting with a given set of parameters were calculated by the following formulation: $L_{total} = \prod_{i=1}^{N}\frac{1}{\sqrt{2\pi\sigma(i)^2}}exp\left[-\frac{\{M(i)-D(i)\}^2}{2\sigma(i)^2}\right]$, where $N$ is the number of data points, $M(i)$, $D(i)$ and $\sigma(i)$ are the value in the model, the value in the experiments, and the variance in the experiments of the *i*-th data points. If the likelihood $L_{total}$ increased from the previous parameters, we adopt the current parameters. Even if the likelihood decreased, we adopt the current parameters with the probability of current $L_{total}$ divided by the previous $L_{total}$. This step was repeated 100,000 steps. We adopted the parameter set giving the highest likelihood as the best fit parameters. The codes are available on our Github (https://github.com/akkimura/pombePheromoneSignalling2025_250423f).

## Supporting information

**S1 Fig. Plate mating assay and validation of the use of commercially available anti-GFP and anti-phospho-MAPK specific antibodies.** (A) A graphical representation of the plate mating assay that induces highly synchronous mating/sexual differentiation as stated in the Materials and Methods "Plate mating assay for synchronous mating" section. (B)

Schematic outlining of the principal of a phospho-specific MAPK antibody, which specifically recognize the phosphorylated TEY motif (pTEpY) of MAPK. (C) Sequence alignment between mammalian ERK1, ERK2 and *S.pombe* MAPK$^{Spk1}$ at the region of the dual phosphorylation required for activation and surrounding residues. Conserved residues are presented in light green, and TEY is highlighted in orange-green. (D)-(F) A Western blot of a time-course of a WT (KT301) and a MAPK$^{Spk1}$-GFP strain (KT3082) over 24 hours, incubated with anti-GFP, anti-pTEpY phospho-ERK and anti-α-tubulin antibodies. The Li-cor Odyssey system was used to detect the signals. (D) The signals obtained by the 700 nm wavelength scan to detect signals from a primary monoclonal mouse anti-GFP antibody ((0.4mg/ml), Roche Cat No 11814460001. Used at a 1:2000 dilution) and the primary anti-α-tubulin antibody TAT1 (generous gift from Keith Gull, 1:3000), followed by the IRDye 680LT secondary antibody (goat anti-mouse antibody, Li-cor 926–32211(1.0 mg/ml), 1:16,000 dilution). α-tubulin was used as an internal loading control in this instance. (E) The exact same blot as in (D) but showing the 800 nm wavelength scan to detect signals from an anti-pTEpY phospho-ERK antibody (#4370 Cell Signaling Technology. Used at 1:2000 dilution) visualized by the IRDye 800CW goat anti-rabbit secondary antibody (Li-cor 926–68020 (1.0 mg/ml), 1:16000 dilution). (F) An overlay image of the 700 and 800 nm channels that are presented in (D) and (E). (G) The phospho-ERK antibody is phospho-specific. Cell extracts were prepared from the time-course of MAPK$^{Spk1}$-GFP tagged WT (KT3082) and *mapkk$^{byr1}$*Δ strains (KT4300), and MAPK$^{Spk1}$-GFP was detected by anti-GFP (red) while phospho-MAPK$^{Spk1}$ was detected by anti-pTEpY phosphor ERK antibody (green). In the *mapkk$^{byr1}$*Δ strain, phosphorylated MAPK$^{Spk1}$-GFP signal is missing, although MAPK$^{Spk1}$-GFP was detectable at 8, 16 and 24 hours after induction of mating. Note that bands indicated by a single (*) and double (**) asterisks were concluded to be irrelevant to phosphor-MAPK$^{Spk1}$-GFP because they do not exactly overlap with the MAPK$^{Spk1}$-GFP signal (in red) in (F) and (G). Furthermore, these bands exist in non-tagged MAPK$^{Spk1}$ strain seen in (E) and in *mapkk$^{byr1}$*Δ strains in (G), further supporting that the bands are irrelevant to phospho-MAPK$^{Spk1}$-GFP. Considering the molecular weight, the double asterisk band may correspond to another MAPK, MAPK$^{Pmk1}$.
(EPS)

**S2 Fig. Original Western blotting images of WT, *byr1.DD*, *ras1.G17V*, *ras1.GV byr1.DD* and *scd1*Δ mutants used to quantify the ppMAPK$^{Spk1}$-GFP levels during the mating process in** Figs 1**,** 2 **and** 3**.** MAPK$^{Spk1}$-GFP Cells were induced for sexual differentiation by the plate mating assay system as described in S1A Fig and the Materials and Methods. Three biological replicates presented were used for quantitation to obtain the mean value and SEM. Actin was used as a loading control (detected by Life Technologies MA1–744; mouse monoclonal RRID:AB 2223496, 1/2000 dilution), and quantitation was carried out using Image Studio ver2.1 software (Licor Odyssey CLx Scanner). **pp**MAPK$^{Spk1}$-GFP was detected by anti-phospho-ERK antibody (#4370 Cell Signaling Technology. Used at 1:2000 dilution) and total MAPK$^{Spk1}$-GFP-2xFLAG protein level was detected using anti-FLAG antibody (Sigma F1804, 1 μg/ul, 1/2000 ditution). The green asterisk indicates the background band, as seen in S1 Fig. Numbers indicate hours after induction of mating.
(EPS)

**S3 Fig. Quantification result of each Western blotting membrane of WT, *byr1.DD*, *ras1.G17V*, *ras1.GV byr1.DD* and *scd1*Δ mutants, presented in** S2 Fig**.** (A) and (B). Relative levels of **pp**MAPK$^{Spk1}$-GFP-2xFLAG (A) and total MAPK$^{Spk1}$-GFP-2xFLAG (B) were quantified for each of the Western blotting membranes presented in S2 Fig. Three biological replicates were made, and each biological replicate data is presented in blue, orange or grey. Quantitation was carried out using Image Studio ver2.1 software (Licor Odyssey CLx Scanner). The Y axis represents relative signal intensities in an arbitrary unit, which is set to the same scale for all the membranes. Numbers of the X-axis indicate hours after induction of mating. For the **pp**MAPK$^{Spk1}$-GFP-2xFLAG measurement, the mean values ±SEM of the three biological replicates are presented in Figs 1C, 1E, 1G, 2A and 3B. For total MAPK$^{Spk1}$-GFP-2xFLAG measurement, the mean values ±SEM of the three biological replicates are presented in the lower row of S3B Fig. (C) Relative ratios of **pp**MAPK$^{Spk1}$/total MAPK$^{Spk1}$ are deduced from each biological replicate and plotted. (D) Fitted values (and 95% confidence intervals) for the **pp**MAPK$^{Spk1}$

relative levels, total MAPK<sup>Spk1</sup> relative levels, and the ratios of **pp**Spk1/total Spk1 levels of the five strains under Generalized Additive Mixed Models (GAMM) are presented. The GAMMs indicate that the initial increase of the **pp**MAPK-<sup>Spk1</sup> occurs distinctively earlier in the *ras1.G17V* mutant than in the wildtype strain. The levels of total MAPK<sup>Spk1</sup> and the **pp**MAPK<sup>Spk1</sup> in the *byr1.DD ras1.G17V* double mutant also increase distinctively earlier than these levels in the *byr1.DD* single mutant. The GAMMs also show that the **pp**Spk1/total Spk1 ratios increase during the first couple of hours after induction of mating in cells that do not carry the *byr1.DD* mutation. Among these cells, the increase rate in the *ras1.G17V* was distinctively higher than in the wildtype cells. Interestingly, the **pp**Spk1/total Spk1 ratio in the *ras1.G17V* cells quickly dropped to become comparable to the wildtype cells, indicating a robust damping mechanism. In contrast, in the presence of the *byr1.DD* mutant, the **pp**Spk1/total Spk1 ratios stayed almost constant throughout the mating process.
(EPS)

**S4 Fig. MAPK<sup>Spk1</sup>-GFP signal intensity and localisation during the wildtype sexual differentiation process.** (A) MAPK<sup>Spk1</sup>-GFP signal intensity and the localisation changes during the wildtype sexual differentiation process. Wildtype cells expressing MAPK<sup>Spk1</sup>-GFP, together with Smd2-tdTomato (KT5951), a nuclear marker, were induced for sexual differentiation by the plate mating assay system as described in the Materials and Methods and S1A Fig for 8–10 hours to collect the data presented in this figure. MAPK<sup>Spk1</sup>-GFP and Smd2-tdTomato signals were captured by a Zeiss LSM 980 Airyscan 2 microscope in Airyscan multiplex-4Y mode. Each image comprises 47 serial images with 0.15 μm intervals along the Z axis taken to span the full thickness of the cell. The presented image is a single Z slice and contains cells and zygotes undergoing shmoo formation and mating (indicated by yellow arrows), karyogamy (indicated by red arrows), horsetail movement (indicated by green arrows) and meosis II (indicated by a blue arrow). (B) The set of serial Z images was used to conduct 3D image segmentation to generate "Segment-Nucleus" and "Segment Total", and the sum of the Spk1-GFP signal intensity in each segment was measured by Imaris software as described in Materials and Methods. (C) Cells and zygotes were classified as "ellipse/oblong", "mating", "karyogamy", "horsetail", "meiosis I (MI)" and "meiosis II (MII)" according to their morphology as illustrated in Panel A and the sum of the nuclear GFP signal, sum of the total GFP signal, and the ratio of nuclear:total GFP signal were measured and analysed by one-way ANOVA followed by post-hoc Tukey's multiple comparisons test. The result is presented as violin plots with a line indicating the median. 20 – 100 cells/zygotes were counted for each category. **** $p \leq 0.0001$, ** $p \leq 0.01$, *$p \leq 0.05$, ns $p > 0.05$. The scale bar represents 10 μm.
(EPS)

**S5 Fig. MAPK<sup>Spk1</sup>-GFP signal intensity and shmoo tip localisation during the wildtype mating process.** (A) MAPK<sup>Spk1</sup>-GFP appears on the cell cortex, in cytoplasm and in the nucleus. Wildtype cells expressing MAPK<sup>Spk1</sup>-GFP, together with Smd2-tdTomato (KT5951), a nuclear marker, were induced for sexual differentiation by the plate mating assay system. MAPK<sup>Spk1</sup>-GFP and Smd2-tdTomato signals were captured at the indicated times after induction of sexual differentiation. Images were taken by a 2D-array scanning laser confocal microscope (Infinity 3, VisiTech) on a NikonTi-E microscope stand equipped with a Hamamatsu Flash 4.0V2 sCMOS camera and a Plan Apo 100x/1.45 objective by spanning the whole thickness of the cells using 41 Z slices with a step size of 200 nm. They were deconvolved using Huygens Essential (Scientific Volume Imaging) and Z projected (maximum intensity) and presented. MAPK<sup>Spk1</sup>-GFP signal was increased after induction of sexual differentiation, and accumulation in the nucleus was observed. Yellow arrows indicate a pair of mating cells, which are presented in (B) with a magnified format to reveal cortical accumulation of MAPK<sup>Spk1</sup>-GFP at the shmoo tips. The scale bar represents 10 μm. (B) Individual serial Z-images of the MAPK<sup>Spk1</sup>-GFP signal found in the pair of mating cells indicated by yellow arrows in (A) are presented in a heat-map format by applying the ImageJ Lookup table"Fire". Yellow arrows indicate a high-intensity accumulation of MAPK<sup>Spk1</sup>-GFP at the shmoo tips. The scale bar represents 5 μm.
(EPS)

**S6 Fig. Cell fusion is prerequisite for MAPK<sup>Spk1</sup> downregulation.** Wildtype cells (KT3082) and the *fus1Δ* cells (KT3982) were induced for sexual differentiation using the plate mating assay system as described in the materials and methods, and cell morphology, MAPK<sup>Spk1</sup>-GFP signals and MAPK<sup>Spk1</sup> phosphorylation status were examined. (A) Cell images 12 hours after induction of sexual differentiation. Images were taken in the same way as in Fig 1. Scale bars: 10 μm. The *fus1Δ* cells exhibit a fusion-deficient phenotype, with accumulation of MAPK<sup>Spk1</sup>-GFP signals, particularly at the cell cortex and in the nucleus. (B) and (C) Quantified **pp**MAPK<sup>Spk1</sup> signal (arbitrary unit) (B) and total MAPK<sup>Spk1</sup> signal (arbitrary unit) (C). Original Western blotting membrane images are presented in (D). Three biological replicates were used for quantitation (error bars are ± SD). Actin was used as a loading control, and quantification was performed using Image Studio ver2.1 software (Licor Odyssey CLx Scanner).
(EPS)

**S7 Fig. *MAPKK<sup>byr1.DD</sup>* mutation causes constitutive activation of MAPK<sup>Spk1</sup>.** The *MAPKK<sup>byr1.DD</sup>* mutant cells (KT3435) were induced for sexual differentiation by the plate mating assay system as described in S1A Fig and the Materials and Methods. Western blotting of **pp**MAPK<sup>Spk1</sup>, total MAPK<sup>Spk</sup>1 and α-tubulin was conducted. The α -tubulin signals were used as an internal control to estimate relative **pp**MAPK<sup>Spk1</sup> levels. (A) The quantitated results obtained from three biological replicates. The mean value for each time point is presented. Error bars represent SD. (B) Original membrane image of the three biological replicates used for the quantitation.
(EPS)

**S8 Fig. MAPK<sup>Spk1</sup>-GFP signal intensity is decreased in the *ras1.G17V* mutant at 16 hours after induction of sexual differentiation.** The MAPK<sup>Spk1</sup>-GFP signals were compared 16 hours after induction of sexual differentiation between the cells of *MAPKK<sup>byr1.DD</sup>* (KT3435) (A) and the cells of *ras1.G17V* (KT3084) (B). Images were taken by a 2D-array scanning laser confocal microscope (Infinity 3, VisiTech) on a NikonTi-E microscope stand equipped with a Hamamatsu Flash 4.0V2 sCMOS camera and a Plan Apo 100x/1.45 objective by spanning the whole thickness of the cells using 31 Z slices with a step size of 200 μm. They were deconvolved using Huygens Essential (Scientific Volume Imaging). The serial Z images were Z-projected using the Image J "Sum slices" to compare the total level of MAPK<sup>Spk1</sup>-GFP signal. Signal intensities were indicated in a heatmap format by applying the ImageJ Lookup table (LUT) "Fire" to the Z-projected images. The LUT gradient is indicated on the right-hand side of the panels. Yellow arrows in Panel A indicate the "paired" cells of the *MAPKK<sup>byr1.DD</sup>* mutant. White arrows in Panel B indicate the "elongated" cells of the *ras1.G17V* mutant. Cells indicated by arrows with a green circle are presented as the original serial Z images in Panel (D). The scale bar represents 10μm. (C) Distribution histograms of pixel signal intensities for *MAPKK<sup>byr1.DD</sup>* (KT3435) (yellow) and *ras1.G17V* (KT3084) (purple). Images that covered 85 cells were subject to ImageJ analysis to measure the MAPK<sup>Spk1</sup>-GFP signal intensity of each pixel of these images. The result is presented as distribution histograms. The peak distribution of pixels of *ras1.G17V* images had a lower signal intensity compared to the one for images of *MAPKK<sup>byr1.DD</sup>*. B.G. indicates the intensities of pixels corresponding to the background area. (D) Cortical accumulation of MAPK<sup>Spk1</sup>-GFP is weaker in the *ras1.G17V* mutant compared to the *MAPKK<sup>byr1.DD</sup>* mutant at 16 hours after induction of sexual differentiation. MAPK<sup>Spk1</sup>-GFP signals of the serial Z sections of the *MAPKK<sup>byr1.DD</sup>* and the *ras1.G17V* mutant cells indicated with the green circles in panels A and B are presented. Signal intensities are shown in a heatmap format by applying ImageJ LUT "Fire". The LUT gradient is presented on the right-hand side of the panels. Arrowheads indicate the accumulation of MAPK<sup>Spk1</sup>-GFP signals at the cell tips. Signal intensities at the paired *MAPKK<sup>byr1.DD</sup>* cell tips were significantly higher than the one seen at the shmoo tip of the *ras1.G17V* cell.
(EPS)

**S9 Fig. Cell morphology and localisation of Cdc42<sup>GTP</sup>, indicated by CRIB-GFP signal, during vegetative growth.** (A) Representative CRIB-GFP signal images of vegetatively growing cells of wildtype (KT5077), *ras1Δ* (5107), *ras1.G17V* (KT5082), *rga4Δ* (5551) and *rga4Δ ras1.G17V* (KT5554) are presented. The scale bar is 10 μm. (B) Quantitated CRIB-GFP signals on the cell cortex of cells presented in (A). The intensity of the GFP signal on the cell cortex was measured

along one of the cell tips with a stronger GFP signal as indicated as a magenta dotted line in the example image on the right (Scale bar: 10 μm) as stated in the Materials and Methods. 40 cells without septum were measured for each strain, and the average curve from all aligned traces per strain was calculated and displayed with respective standard error of the mean curves (dashed lines) as described in Materials and Methods.

(EPS)

**S10 Fig. Original Western blotting images used to quantify the ppMAPK$^{Spk1}$-GFP and total MAPK$^{Spk1}$-GFP levels during the mating process presented in** Fig 3**.** (A) Original Western blotting membranes used to quantify the **pp**MAPK$^{Spk1}$-GFP and total MAPK$^{Spk1}$-GFP to generate the graphs presented in Figs 3D and S10D. Strains used were; scd1Δ (KT4061), scd1Δ ras1.G17V (KT4056) and scd1Δ MAPKK$^{byr1.DD}$ (KT4047). (B) Original Western blotting membranes used to quantify the **pp**MAPK$^{Spk1}$-GFP and total MAPK$^{Spk1}$-GFP to generate the graphs presented in Figs 3E and S10E. Strains used were; ras1Δ (KT4323), MAPKK$^{byr1.DD}$ (KT3435) and ras1Δ MAPKK$^{byr1.DD}$ (KT4359) (C) Original Western blotting membranes used to quantify the **pp**MAPK$^{Spk1}$-GFP and total MAPK$^{Spk1}$-GFP to generate the graphs presented in Figs 3F and S10F. Strains used were; in mapkkk$^{byr2}$Δ (KT3763), MAPKK$^{byr1.DD}$ (KT3435) and mapkkk$^{byr2}$Δ MAPKK$^{byr1.DD}$ (KT4010). (D)-(F) Quantified results of three biological replicates (error bars are ± SEM) are presented for total MAPK$^{Spk1}$-GFP. Original membranes are presented in the panels (A)-(C). Indicated mutant cells were induced for sexual differentiation by the plate mating assay system as described in the Materials and Methods. Western blotting of ppMAPK$^{Spk1}$, total MAPK$^{Spk1}$ and α-tubulin was conducted in the same way as S6 Fig. The green asterisks indicate the same background band recognized by the anti-phospho-ERK antibody (#4370 Cell Signaling Technology. Used at 1:2000 dilution) seen in S1 Fig. α-tubulin was used as a loading control, and quantitation was performed using Image Studio ver2.1 software (Licor Odyssey CLx Scanner).

(EPS)

**S11 Fig. Original Western blotting membranes used in** Fig 5A**.** Original Western blotting membranes used to quantify ppMAPK$^{Spk1}$-GFP to generate the graph in Fig 5A. Cells harboring ras1.G17V and MAPK$^{Spk1}$-GFP-2xFLAG (KT5940) were transformed with either pRep1 empty vector, pRep1-scd1(760–872)-2xFLAG or pRep1-byr2(65–180)-2xFLAG and cultured in MM + N without thiamine for 24 hours. Cells were induced for sexual differentiation by the plate mating assay system, and the levels of ppMAPK$^{Spk1}$-GFP, total MAPK$^{Spk1}$ and α-tubulin (loading control) were examined by Western blotting. Three biological replicates are shown. The quantitation was carried out using Image Studio ver2.1 software (Licor Odyssey CLx Scanner). The expressions of Scd1(760–872)-2xFLAG and Byr2(65–180)-2xFLAG were confirmed by anti-FLAG antibody (Sigma F1804, 1 μg/ul, 1/2000 dilution).

(EPS)

**S12 Fig. The expression of Scd1-PB1 or Byr2-RBD affects cell morphology of vegetatively growing cells.** Cell morphology and localisation of Cdc42$^{GTP}$, indicated by the CRIB-GFP signal, of vegetatively growing cells were compromised by the expression of Scd1-PB1 or Byr2-RBD. Cells harbouring ras1.G17V and CRIB-GFP (KT5938) were transformed with plasmids carrying either scd1-PB1 or byr2-RBD, and live cell images of the CRIB-GFP signals were taken after cultured in MM + N without thiamine for 24 hours. Overexpression of Scd1-PB1 had a more profound effect than that of Byr2-RBD. The scale bars are 10 μm.

(EPS)

**S13 Fig. Comparable signal intensities of Scd1-GFP and GFP-Byr2 at the mating shmoo tip.** (A) Brightfield and GFP images of cells harbouring byr2-GFP (KT5032). Cells were imaged 24 hours after induction for sexual differentiation. No zygotes were observed, demonstrating that C-terminal GFP tagging of Byr2 made cells sterile. (B) and (C) Brightfield and GFP images of Cells harbouring scd1-GFP (KT4382) and GFP-byr2 (KT5135). Cells were imaged in rich media (YE) (B) or induced for sexual differentiation for 8 hours (C). Images were taken with a Zeiss LSM 980 Airyscan 2 microscope in

LSM confocal mode. 32 serial Z-sections were captured with 0.24 μm intervals. For brightfield images, a single Z-slice is shown; for GFP images, Z-projected (maximum intensity) images are shown. Scale bars are 10 μm.
(EPS)

**S14 Fig. Original Western blotting images used to quantify the ppMAPK$^{Spk1}$-GFP and total MAPK$^{Spk1}$-GFP levels during the mating process in** Fig 6. (A) Original Western blotting membranes used to quantify the **pp**MAPK$^{Spk1}$-GFP and total MAPK$^{Spk1}$-GFP to generate the graphs presented in Figs 6A and S14C. Strains used were; *ste4Δ* (KT4376), *ste4Δ ras1.G17V* (KT5143) and *ste4Δ MAPKK$^{byr1.DD}$* (KT5136). (B) Original Western blotting membranes used to quantify the **pp**MAPK$^{Spk1}$-GFP and total MAPK$^{Spk1}$-GFP to generate the graphs presented in Figs 6C and S14D. Strains used were; *ste6Δ* (KT4333), *ste6Δ ras1.G17V* (KT4998) and *ste6Δ MAPKK$^{byr1.DD}$* (KT5139). (C) and (D) Quantified results of three biological replicates (error bars are±SEM) are presented for total MAPK$^{Spk1}$-GFP. Original membranes are presented in the panels (A) and (B). Indicated mutant cells were induced for mating by the plate mating assay system as described in the Materials and Methods. Western blotting of ppMAPK$^{Spk1}$, total MAPK$^{Spk1}$ and α-tubulin was conducted in the same way as S6 Fig. The green asterisks indicate the same background band recognized by the anti-phospho-ERK antibody (#4370 Cell Signaling Technology. Used at 1:2000 dilution) seen in S1 Fig. α-tubulin was used as a loading control, and quantitation was carried out using Image Studio ver2.1 software (Licor Odyssey CLx Scanner).
(EPS)

**S15 Fig. Original Western blotting membranes used in** Fig 7. (A) Original Western blotting membranes used to quantitate ppMAPK$^{Spk1}$-GFP to generate the graph in Fig 7A. Strains used were; *gpa1Δ* (KT4335), *gpa1Δ ras1.G17V* (KT5023), *gpa1Δ MAPKK$^{byr1.DD}$* (KT4353) and *gpa1Δ ras1.val17 MAPKK$^{byr1.DD}$* (KT5035). (B) Original Western blotting membranes used to quantitate ppMAPK$^{Spk1}$-GFP to generate the graph in Fig 7C. Strains used were; *h$^-$* WT (KT4190), *h$^-$ gpa1.QL* (KT5059), *h$^-$ ras1.G17V* (KT4233), *h$^-$ gpa1.QL ras1Δ* (KT5070), h$^-$ *MAPKK$^{byr1.DD}$* (KT4194). The green asterisk indicates a background band recognized by anti-phospho-ERK antibody (#4370 Cell Signaling Technology. Used at 1:2000 dilution). α-tubulin was used as a loading control and quantitation was carried out using Image Studio ver2.1 software (Licor Odyssey CLx Scanner).
(EPS)

**S16 Fig. Components and framework of the mathematical model.** Components and framework of the mathematical model (Model C in this instance) of wildtype and signalling mutants, *ras1.G17V*, *MAPKK$^{byr1.DD}$*, *ras1.G17V MAPKK$^{byr1.DD}$* and *scd1Δ*. Changes corresponding to each mutant are indicated as follows: Grey: removed components or interactions, orange: the activation process does not require the upstream elements. For the exact implementation of the mutants, see Materials and Method. The measured components, total MAPK$^{Spk1}$, **pp**MAPK$^{Spk1}$ and Cdc42$^{GTP}$, are indicated in purple, green and red, respectively.
(EPS)

**S17 Fig. Model A and its fittings for the total MAPK$^{Spk1}$, ppMAPK$^{Spk1}$, and active Cdc42 levels.** (A) Model A components and framework. For the detailed implementation of the mutants, see S16 Fig and Materials and Methods. The measured components, total MAPK$^{Spk1}$, **pp**MAPK$^{Spk1}$ and Cdc42$^{GTP}$, are shown in purple, green and red, respectively. (B) and (C) Model A fittings for the total MAPK$^{Spk1}$ (B) and **pp**MAPK$^{Spk1}$ (C) levels in wildtype, *ras1.G17V*, *MAPKK$^{byr1.DD}$*, *ras1. G17V MAPKK$^{byr1.DD}$* and *Cdc42GEF$^{scd1}$Δ* mutants. Mean values (open circles) and SD values (error bars) of the experimental results are shown in pale colours, and the model-fitted values are shown as filled circles in darker colours. (D) Model A fitting of the active Cdc42 levels in wildtype and *ras1.G17V* mutants. Mean values (open circles) and SD values (error bars) of the experimental results are shown in pale colours, and the model-fitted values are shown in filled circles in darker colours.
(EPS)

**S18 Fig. Model B fittings for the total MAPK<sup>Spk1</sup> levels.** Model B fittings for the total MAPK<sup>Spk1</sup> levels in wildtype, *ras1. G17V*, *MAPKK<sup>byr1.DD</sup>*, *ras1.G17V MAPKK<sup>byr1.DD</sup>* and *Cdc42GEF<sup>scd1</sup>Δ* mutants. Mean values (open circles) and SD values (error bars) of the experimental results are shown in pale colours, and the model-fitted values are shown as filled circles in darker colours.
(EPS)

**S19 Fig. Model C and its fittings for the total MAPK<sup>Spk1</sup>, ppMAPK<sup>Spk1</sup>, and active Cdc42 levels.** (A) Model C components and framework. For the detailed implementation of the mutants, see S16 Fig and Materials and Methods. The measured components, total MAPK<sup>Spk1</sup>, **pp**MAPK<sup>Spk1</sup> and Cdc42<sup>GTP</sup>, are shown in purple, green and red, respectively. (B) and (C) Model C fittings for the total MAPK<sup>Spk1</sup> (B) and **pp**MAPK<sup>Spk1</sup> (C) levels in wildtype, *ras1.G17V*, *MAPKK<sup>byr1.DD</sup>*, *ras1.G17V MAPKK<sup>byr1.DD</sup>* and *Cdc42GEF<sup>scd1</sup>Δ* mutants. Mean values (open circles) and SD values (error bars) of the experimental results are shown in pale colours, and the model-fitted values are shown as filled circles in darker colours. (D) Model C fitting of the active Cdc42 levels in wildtype and *ras1.G17V* mutants. Mean values (open circles) and SD values (error bars) of the experimental results are shown in pale colours, and the model-fitted values are shown in filled circles in darker colours.
(EPS)

**S20 Fig. Prediction ability of Models B and C.** Model B and C were used to predict the experimental results shown in Figs 3E, 3F, 6A, 6C, 7A and 7C, involving 20 fission yeast strains. First, the **pp**Spk1 of *MAPKK<sup>byr1Δ</sup>* (S1F Fig) was predicted by fixing the amount of MAPKK<sup>Byr1</sup> to zero in Model B/C. The models predicted a minimum ppSpk1 level in *MAPKK<sup>byr1Δ</sup>*, consistent with the experiment.**Fig 3E** prediction involving *MAPKK<sup>byr1.DD</sup>*, *ras1Δ*, and *ras1Δ MAPKK<sup>byr1.DD</sup>*: the amount of Ras1 was fixed to zero for *ras1Δ*. Model B predicted the **pp**MAPK<sup>Spk1</sup> relative levels in these three strains both qualitatively and quantitatively well. Model C also made a qualitatively good prediction, but it predicted the *MAPKK<sup>byr1.DD</sup>* single mutation to generate a higher level of **pp**MAPK<sup>Spk1</sup> compared to the *ras1Δ MAPKK<sup>byr1.DD</sup>* double mutant, although the experimental result did not show a significant difference between these two strains. This discrepancy was because that Model C was constructed on an assumption that Ras1 positively contributes to the **pp**Spk1 production independent of Byr1 at the rate *k27* (to cause an earlier increase of the **pp**Spk1 in the *ras1.G17V MAPKK<sup>byr1.DD</sup>* double mutant compared to the *MAPKK<sup>byr1.DD</sup>* single mutant as suggested in S3D Fig). The Model C result suggests that the predicted Ras1 contribution to the MAPK<sup>Spk1</sup> activation is more complex than we considered in the proposed models.**Fig 3F** prediction involving *MAPKK<sup>byr1.DD</sup>*, *mapkkk<sup>byr2</sup>Δ MAPKK<sup>byr1.DD</sup>*, and *mapkkk<sup>byr2</sup>Δ*: the activation of MAPKK<sup>Byr1</sup> was set to zero for the strains carrying the *mapkkk<sup>byr2</sup>Δ* allele. Both Model B and C predicted the experimental result qualitatively and quantitively. **Fig 6A** prediction involving *ste4Δ MAPKK<sup>byr1.DD</sup>*, *ste4Δ ras1.G17V*, and *ste4Δ*: the activation of MAPKK<sup>Byr1</sup> was set to zero for the strains carrying the *ste4Δ* allele. Therefore, the *ste4Δ* model should behave identically to the *mapkkk<sup>byr2</sup>Δ* model examined above. Both Model B and C predicted the experimental result qualitatively and quantitively. **Fig 6C** prediction involving *ste6Δ MAPKK<sup>byr1.DD</sup>*, *ste6Δ ras1.G17V*, and *ste6Δ*: the activation of Ras1 was set to zero for the strains carrying the *ste6Δ* allele. Both Model B and C predicted a higher **pp**Spk1 for *ste6Δ MAPKK<sup>by1.DD</sup>*. Model C also predicted a lower ppSpk1 throughout the time course, which agrees with the experimental result. Model B predicts a lower **pp**Spk1 for *ste6Δ ras1<sup>G17V</sup>* and *ste6Δ* for the later time points (10–25 hours after induction of mating), which were consistent with the experiment. However, Model B's prediction of substantial **pp**Spk1 increase in the earlier time points in the *ste6Δ ras1<sup>G17V</sup>* and *ste6Δ* mutants was not seen in the experimental result. Model C predicts a high **pp**Spk1 for the *ste6Δ ras1.G17V*, which is not the case in the experimental result. The result indicates that Ste6 is involved in activating **pp**Spk1 downstream of Ras1, which is not included in our current model.**Fig 7A** prediction involving *gpa1Δ ras1.G17V MAPKK<sup>byr1.DD</sup>* triple mutant, *gpa1Δ ras1.G17V* double mutant, *gpa1Δ MAPKK<sup>byr1.DD</sup>* double mutant, and *gpa1Δ* single mutant: the activation of both Ras1 and Byr1 was set to zero for the strains carrying the *gpa1Δ* allele. Model B predicted the experimental result very well. Model C also reproduced the experimental result of the *gpa1Δ ras1.G17V* double mutant and the *gpa1Δ* single mutant, predicting the minimum level of the **pp**Spk1. Meanwhile, for the

*gpa1Δ ras1.G17V MAPKK^byr1.DD* triple mutant and the *gpa1Δ MAPKK^byr1.DD* double mutant, Model C predicted a higher **pp**Spk1 level for the *gpa1Δ ras1.G17V MAPKK^byr1.DD* triple mutant than the *gpa1Δ MAPKK^byr1.DD* double mutant. This is because Model C postulates that Ras1 can boost the **pp**Spk1 level independent of MAPKK^Byr1 at a rate constant k27, as discussed above. This Ras1 contribution should be smaller than we assumed in Model C. **Fig 7C prediction involving h- MAPKK^byr1.DD, h⁻ gpa1.QL, h⁻ ras1.G17V, h⁻ gpa1.QL ras1Δ and h⁻ wildtype:** in order to reflect the lack of pheromone signalling in these *h⁻* strains, the activation levels of Ras1 and Byr1 were reduced to 1/100 of the *h^90* wild type values, and the amount of active Ste4/Ste6 (i.e., pheromone sensing unit) was set to the value of 0.2. Both Model B and C recapitulated the experimental result qualitatively by predicting relatively higher **pp**Spk1 levels for *h⁻ MAPKK^byr1.DD* and *h⁻ gpa1.QL* strains, whereas the rest of the strains are predicted to produce lower, but non-zero, **pp**Spk1 levels. Model C predicted the *h⁻ ras1.G17V* mutant to produce an initial **pp**Spk1 increase, which is higher than the experiment, indicating that the activity of Ras1.G17V was estimated too high.
(EPS)

**S21 Fig. Loss of *sxa2* compromises the negative feedback in the *ras1. G17V* mutant for MAPK^Spk1 activation.** The *ras1.G17V* cells (KT3084) and the *ras1.G17V sxa2Δ* cells (KT4037) were induced for sexual differentiation using the plate mating assay system as described in the materials and methods, and cell morphology, MAPK^Spk1-GFP signals and MAPK-^Spk1 phosphorylation status were examined. (A) Cell images 4 hours and 8 hours after induction of sexual differentiation. Images were taken in the same way as in Fig 1. Scale bars: 10 μm. Both cells exhibited the "elongated" phenotype. The *ras1.G17V sxa2Δ* cells exhibited morphological changes at an earlier stage (4 h) and their morphology was more heterogenous after a longer incubation time (8h). (B) and (C) Quantified **pp**MAPK^Spk1 signal (arbitrary unit) (B) and total MAPK^Spk1 signal (arbitrary unit) (C). Original Western blotting membrane images are presented in (D). Three biological replicates were used for quantitation (error bars are ± SD). Actin was used as a loading control, and quantification was performed using Image Studio ver2.1 software (Licor Odyssey CLx Scanner).
(EPS)

**S22 Fig. Loss of *sxa2* compromises the negative feedback in the *byr1.DD* mutant for Cdc42 activation.** The *byr1. DD* cells (KT3435) and the *byr1.DD sxa2Δ* cells (KT4031) were induced for sexual differentiation using the plate mating assay system as described in the materials and methods, and cell morphology, MAPK^Spk1-GFP signals and MAPK^Spk1 phosphorylation status were examined. (A) Cell images 4 hours and 8 hours after induction of sexual differentiation. Images were taken in the same way as in Fig 1. Scale bars: 10 μm. The "paired" phenotype of *byr1.DD* single mutant was converted to the "elongated" phenotype that represents hyper-activation of Cdc42. (B) and (C) Quantified **pp**MAPK^Spk1 signal (arbitrary unit) (B) and total MAPK^Spk1 signal (arbitrary unit) (C). Original Western blotting membrane images are presented in (D). Three biological replicates were used for quantitation (error bars are ± SD). Actin was used as a loading control, and quantification was performed using Image Studio ver2.1 software (Licor Odyssey CLx Scanner).
(EPS)

**S1 Table. Table 1.** Model parameters and opimised values.
(XLSX)

**S2 Table. Table 2.** Strains used in this study.
(DOCX)

## Acknowledgments

The authors thank Tatsuya Maeda, Thibault Mayor, Janni Petersen, Louise Fairall, John Schwabe, David Critchley, Andrey Reviyakin, Gary Willars and Mohan Harihar for helpful suggestions, stimulating discussions and critical reading of the manuscript. We thank Dr. Hiroyuki Kubota (Kyushu Univ) for the discussion on the modelling of signalling networks, Drs.

Genta Ueno, Shinya Nakano, Daisuke Murakami, Yusaku Ohkubo (The Institute of Statistical Mathematics) for the discussion on the parameter search methods. We thank PROTEX and PNACL at University of Leicester for their technical assistance. The work was conducted using the Advanced Imaging Facility at the University of Leicester (PRID:SCR_020967) that houses Zeiss LSM 980 Airyscan2 funded by BBSRC (BB/S019510/1). We are grateful to Kazu Shiozaki, Keith Gull and Yeast Genetic Resource Center for providing strains and antibodies.

## Author contributions

**Conceptualization:** Emma J. Kelsall, Akatsuki Kimura, Kayoko Tanaka.

**Data curation:** Akatsuki Kimura.

**Formal analysis:** Akatsuki Kimura, Ábel Vértesy, Edda Klipp.

**Funding acquisition:** Akatsuki Kimura, Ábel Vértesy, Kornelis R. Straatman, Edda Klipp, Kayoko Tanaka.

**Investigation:** Emma J. Kelsall, Akatsuki Kimura, Ábel Vértesy, Kornelis R. Straatman, Mishal Tariq, Raquel Gadea, Chandni Parmar, Gabriele Schreiber, Shubhchintan Randhawa, Takashi Y. Ida, Cyril Dominguez, Kayoko Tanaka.

**Methodology:** Emma J. Kelsall, Akatsuki Kimura, Ábel Vértesy, Kornelis R. Straatman, Takashi Y. Ida, Kayoko Tanaka.

**Software:** Akatsuki Kimura, Ábel Vértesy, Kornelis R. Straatman, Edda Klipp.

**Supervision:** Cyril Dominguez, Edda Klipp, Kayoko Tanaka.

**Validation:** Kayoko Tanaka.

**Visualization:** Kayoko Tanaka.

**Writing – original draft:** Akatsuki Kimura, Kayoko Tanaka.

**Writing – review & editing:** Emma J. Kelsall, Akatsuki Kimura, Ábel Vértesy, Kornelis R. Straatman, Takashi Y. Ida, Cyril Dominguez, Edda Klipp, Kayoko Tanaka.

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
