## [Decision Letter · Decision Letter 0]

2 Sep 2025

PGENETICS-D-25-00520

Constitutively active RAS prolongs Cdc42 signalling, while MAPK signalling is attenuated during fission yeast mating

PLOS Genetics

Dear Dr. Tanaka,

Thank you for submitting your manuscript to PLOS Genetics. The reviewers appreciated you careful analysis and modelling of the mating pathway in fission yeast. However, after careful consideration, we feel that the manuscript does not fully meet PLOS Genetics's publication criteria as it currently stands. Addressing the reviewers comments will require additional experiments, Therefore, we invite you to submit a revised version of the manuscript that addresses the points raised during the review process.

Both reviewers have queried your interpretation of Fig. 1C, which requires attention. In addition, both raised concerns about the quality of some figures. The reviewers have also made suggestions about experimentally testing some of your hypotheses.

Please submit your revised manuscript within 60 days Nov 01 2025 11:59PM. If you will need more time than this to complete your revisions, please reply to this message or contact the journal office at plosgenetics@plos.org. Please include the following items when submitting your revised manuscript:

We look forward to receiving your revised manuscript.

Kind regards,

Geraldine Butler

Section Editor

PLOS Genetics

Eva Stukenbrock

Section Editor

PLOS Genetics

Aimée Dudley

Editor-in-Chief

PLOS Genetics

Anne Goriely

Editor-in-Chief

PLOS Genetics

**Additional Editor Comments:**

Reviewer #1:

Reviewer #2:

**Journal Requirements:**

At this stage, the following Authors/Authors require contributions: Kayoko Tanaka. Please ensure that the full contributions of each author are acknowledged in the "Add/Edit/Remove Authors" section of our submission form.

The list of CRediT author contributions may be found here: https://journals.plos.org/plosgenetics/s/authorship#loc-author-contributions

https://journals.plos.org/plosgenetics/s/submission-guidelines#loc-parts-of-a-submission

5) We have noticed that you have uploaded Supporting Information files, but you have not included a list of legends. Please add a full list of legends for your Supporting Information files after the references list.

Potential Copyright Issues:

i) Figure S1A. Please confirm whether you drew the images / clip-art within the figure panels by hand. If you did not draw the images, please provide (a) a link to the source of the images or icons and their license / terms of use; or (b) written permission from the copyright holder to publish the images or icons under our CC BY 4.0 license. Alternatively, you may replace the images with open source alternatives. See these open source resources you may use to replace images / clip-art:

7) Please amend your detailed Financial Disclosure statement. This is published with the article. It must therefore be completed in full sentences and contain the exact wording you wish to be published.

2) If any authors received a salary from any of your funders, please state which authors and which funders..

8) Kindly revise your competing statement to align with the journal's style guidelines: 'The authors declare that there are no competing interests.'

**Reviewers' comments:**

Reviewer's Responses to Questions

**Comments to the Authors:**

Reviewer #1: Summary:

Small GTPases of the Ras family are key signaling regulators across eukaryotes. During the fission yeast mating, partner cells communicate via pheromones that lead to Ras1 activation, which then propagates onto the cell polarity Cdc42 module and the MAP kinases cascade. Even though majority of regulators in this signaling network has been previously identified, the dynamics of the signaling output is still poorly understood. In the present study, the authors monitor the MAPKSpk1 activation in a series of mutants that hyperactivate or inactivate upstream signaling components. The authors perform extensive amount of experiments that confirms previously suggested network architecture (reviewed in Sieber et al.,2023) and advances the understanding of several components of the signaling network and the overall signaling dynamics. Finally, the authors build a complex mathematical model of this multicomponent system that captures the observed signaling dynamics and suggest unverified roles for additional signaling regulators.

Major points

1. The authors establish a protocol to induce highly synchronous mating (Fig.1C), where the fraction of “mating” increased 0-80% from 7h-15h. However, mating results in fusion and diploid cells rapidly arrest mating, thus the fraction of actively mating cells is actually considerably lower than presented in the graph, which is misleading. Thus, reporting fused cells, cells that are shmooing and cells that are scanning and arrested (Bendezu and Martin, 2013) would be more informative. This, also raises the following questions:

1a. How much of the ppSpk1 drop can be explained by wildtype cells fusing? What is the profile in fus1∆ cells that never fuse?

1b. Does the fact that WT cells fuse and numerous mutants fail to fuse complicate the comparisons between the mutants and controls at late timepoints in the experiment? This should be discussed where relevant.

2. The authors highlight in the abstract that they implement the competition between the Ras1 effectors MAPKKKByr2 and Cdc42-GEFScd1. Two points regarding this are:

2a. The authors convincingly show that Byr2 and Scd1 compete to bind to Ras1-GTP in vitro. While these results show that Byr2 and Scd1 can compete, their protein levels relative to that Ras1-GTP will also be important (both Scd1 and Byr2 are low level proteins and Ras1 is abundant). The authors show in vivo that Ras1-binding domains of Byr2/Scd1 outcompete native proteins but this is done upon extreme overexpression of Byr2/Scd1 from the pREP1 vectors. How much competition of Byr2 and Scd1 there is at native levels thus remains unclear and should be explored further (e.g. changes in gene copy number)

2b. The authors do not discuss the importance of Byr2/Scd1 putative competition in the model nor do they use the model to explore its importance. Also, it is not clearly stated (nor immediately evident to this reviewer) how the model captures the proposed competition.

3. The authors highlight in the abstract that they implement the pheromone signaling negative feedback and explain that this is a critical aspect for the model to capture the observed dynamics (Model B). They propose factors such as Sxa2 and Rgs1 as key regulators of negative feedback, but they do not experimentaly test their mutants regarding ppSpk1, tSpk1 nor CRIB, which would help stregthen the predictions of their model. In particular, the ppSpk1 delay in double mutants with byr1DD would be interesting.

4. In the byr1DDras1G17V background deleting the gpa1 reduces the ability to form elongated cells (compare Fig.7B and 2B). How do authors explain this difference in the context of the signaling network?

5. The micrographs of Spk1-GFP in main figures (e.g. Fig.1, 2, 6,7) is far worse quality that that in the Fig.S4, which produces more convincing images. It is possibly due to the quality of the figures I received. In either case, the quality should be improved and quantifications of the nucleo-cytoplasmic ratio should be provided where relevant.

6. The model should be explained in much more detail in the materials and methods. This should include reasoning and justifications, including supporting publications, for proposed reactions (e.g. k15). Have any of the constants been mesured before, including other yeast or in vitro systems? Would any of such measurments restrict the model given that some k (e.g k14, k15, k6) values vary by orders of magnitude between models?

7. Quantifications of the tSpk1 should accompany those of ppSpk1. This is particulary important in instances where the mutants severely downregulate tSpk1(Fig.3D,E ; 6A,C), which should be highlighted and explained in the text also.

Minor points

1. The manuscript is very dense and difficult to read, and more clarity would be appreciated. The authors present the work around the constitutively active Ras, but they study more globally how fission yeast pheromone signaling pathway functions. This results in initial confusion/misleading of the reader. For example, the reason for MAPK singaling attenuation is due to upstream, Ras1-independent regulation, which becomes clear only late in the manuscript.

2. How do authors explain that overexpression of scd1-PB1 and byr2-RBD has little to no effect on tSpk1 but a strong effect on ppSpk1?

3. Introduce the axis for the mating in Fig.1C.

4. Axis labels missing in Fig.3D, 3E, 6A, 6C, 7A, 7C.

5. Why is the profile of ppSpk1/tSpk1 different in Fig5A(right pannel, pREP1) and Fig.S3C (ras1G17V)?

6. A better definition and qunatification of the “fus” vs. “elongated” phenotypes would be beneficial.

7. Fig. S12/s14 say Model B in graphs, but presumably it’s models A/C.

Reviewer #2: Reviewer Comments to the Authors:

This manuscript by Kelsall et al. investigates two key aspects of the yeast mating pathway: the pheromone-induced MAPK signalling cascade and the morphological changes leading to shmoo formation and cell-cell pairing. Although the major components of this pathway and their roles in mating are well characterized, and some of the findings reported here overlap with previous studies, the authors employ a valuable approach by analyzing synchronous mating cultures and tracking MAPK pathway activation over time. Specifically, they monitor the phosphorylation status of the MAPK Spk1 via western blot analysis.

The central finding, supported by mathematical modeling, is that constitutive activation of Ras1, a small GTPase positioned upstream of both the MAPK cascade and the Cdc42-mediated morphological response, does not hyperactivate the pheromone response. Instead, it primarily affects Cdc42 activation, thereby promoting the formation of elongated shmoos.

Overall, I consider the manuscript a meaningful contribution to the field. It offers new insights into a well-characterized signalling pathway through a novel experimental design. I support publication in PLOS Genetics pending revisions. My specific comments are provided below:

- “Gpa1, the α-subunit of the pheromone receptor-coupled G-protein, relays the pheromone signal into the cell through Ras1, leading to the activation of the MAPK cascade” I recommend narrowing this statement. While it is known that the Ras1-GEF Ste6 is activated downstream of Gpa1 and the pheromone–MAPK signalling, MAPK activation also involves inputs that are independent of Ras1.

- Figure 1C: Regarding WT mating efficiency over time - does the percentage refer to mating pairs or zygotes? Please clarify what is represented. Also, the y-axis on the left is labeled as A.U. (arbitrary units), to quantify the pSpk1 signal. Can you specify what the percentage is?

- Synchronous mating assay: How does the mating efficiency in your assay compare to previously established methods for synchronous mating?

- "Signal was first detected three hours after induction of mating and reached its peak at about five to seven hours, when cell fusion was also initially observed" How is cell fusion monitored in your assay? Please specify the method used.

- Regarding western blots in supplementary figures, the authors use actin as a loading control in some blots and tubulin in others. Is there a specific reason for switching between these controls? Additionally, in some cases the loading controls appear weak or inconsistent, making interpretation of the blots less straightforward, even though the quantification presented in the graphs is relatively clear.

- Although you measure Spk1 levels and phosphorylation via western blot, it would be informative to quantify Spk1-GFP levels from microscopy images, at least in WT, ras1-G17V, and byr1-DD cells. For instance, based on the images, Spk1-GFP expression does not appear to decrease significantly in ras1-G17V cells. Have you considered quantifying the nuclear-to-cytoplasmic ratio of the signal? This could provide additional insights into Spk1 localization and activation.

- CRIB-GFP signal in ras1Δ cells: The statement, “During the vegetative cycle, ras1Δ cells show spherical morphology where polarised localisation of active Cdc42 is compromised,” is accurate but was also previously demonstrated and quantified in Lamas et al., 2020 using CRIB-GFP localization at the cell ends.

- Some of the fluorescence microscopy images appear out of focus or blurry. Would it be possible to replace these with higher-quality images to improve data presentation?

- Figure 3: Please include quantification of the pSpk1 signal in scd1Δ ras1-G17V and scd1Δ byr1-DD strains.

- Have you examined the effects of expressing byr2-RBD or scd1-PB1 during mitosis? Are there any morphological phenotypes associated with their expression at this stage?

- Line 238: This should read rga4, not rgl4.

- Line 241: While the point is valid, it has been previously shown that Ras activity, together with scaffold-mediated positive feedback, promotes Cdc42 activation.

- Figure S12: The label “Model B” in the graph should be corrected to “Model A.”

- Line 438: The correct notation should be Ras1-G17V.

**Have all data underlying the figures and results presented in the manuscript been provided?**

Large-scale datasets should be made available via a public repository as described in the *PLOS Genetics*
data availability policy, and numerical data that underlies graphs or summary statistics should be provided in spreadsheet form as supporting information., and numerical data that underlies graphs or summary statistics should be provided in spreadsheet form as supporting information., and numerical data that underlies graphs or summary statistics should be provided in spreadsheet form as supporting information., and numerical data that underlies graphs or summary statistics should be provided in spreadsheet form as supporting information.

Reviewer #1: Yes

Reviewer #2: Yes

PLOS authors have the option to publish the peer review history of their article (what does this mean?). If published, this will include your full peer review and any attached files.). If published, this will include your full peer review and any attached files.). If published, this will include your full peer review and any attached files.). If published, this will include your full peer review and any attached files.

...

Reviewer #1: No

Reviewer #2: No

**Figure resubmission:**
---

## [Decision Letter · Decision Letter 1]

8 Mar 2026

PGENETICS-D-25-00520R1

Constitutively active RAS prolongs Cdc42 signalling, while MAPK signalling is attenuated during fission yeast mating

PLOS Genetics

Dear Dr. Tanaka,

Thank you for submitting your revised manuscript to PLOS Genetics. The reviewers agree that you have addressed most of their concerns, and overall, the manuscript is acceptable for publication. However, I encourage you to address one remaining concern from reviewer 1, that "you should explicitly that the extent of Scd1 Byr2 competition in vivo remains to be seen" and that the discussion should reflect that.  Therefore, we invite you to submit a slightly revised version of the manuscript.

Please submit your revised manuscript within by Apr 07 2026 11:59PM. If you will need more time than this to complete your revisions, please reply to this message or contact the journal office at plosgenetics@plos.org. Please include the following items when submitting your revised manuscript:

We look forward to receiving your revised manuscript.

Kind regards,

Geraldine Butler

Section Editor

PLOS Genetics

Eva Stukenbrock

%CORR_ED_EDITOR_ROLE%

PLOS Genetics

Aimée Dudley

Editor-in-Chief

PLOS Genetics

Anne Goriely

Editor-in-Chief

PLOS Genetics

**Journal Requirements:**

1) Please provide an Author Summary. This should appear in your manuscript between the Abstract (if applicable) and the Introduction, and should be 150-200 words long. The aim should be to make your findings accessible to a wide audience that includes both scientists and non-scientists. Sample summaries can be found on our website under Submission Guidelines:

https://journals.plos.org/plosgenetics/s/submission-guidelines#loc-parts-of-a-submission

4) Please amend your detailed Financial Disclosure statement. This is published with the article. It must therefore be completed in full sentences and contain the exact wording you wish to be published.

2) If any authors received a salary from any of your funders, please state which authors and which funders..

**Reviewers' comments:**

Reviewer's Responses to Questions

**Comments to the Authors:**

Reviewer #1: I thank the authors for their efforts in addressing most of the points from the initial review. Thus, I would like to recommend publishing the article.

However, I would also like to note that it remains unclear how much competition of Byr2 and Scd1 for Ras1-GTP there is at native protein levels (point 2a). The new imaging data suggests “that these two molecules are available at the fusion site at comparable levels”. This also suggests that having used the pREP1 vectors (nmt1 protein is 57312.32 molecules/cells) makes it hard to strengthen the idea that Scd1 (453 molecules/cell) and Byr2 (unknown molecules/cell, suggestedly comparable to Scd1) compete for Ras1 (7003.3 molecules/cell for total Ras1 protein, which is order of magnitude greater than either Scd1 or Byr2). For this reason, I would like to recommend to further tone down their conclusions and state explicitly that the extent of Scd1 Byr2 competition in vivo remains to be seen. This is not to take away from the solid biochemistry work in the current manuscript; quite to the contrary, it is to encourage the difficult but important future work on addressing the Scd1 Byr2 competition in vivo.

Reviewer #2: The reviewer is satisfied with the authors' revisions to the text and figures and supports the manuscript for publication.

**Have all data underlying the figures and results presented in the manuscript been provided?**

Large-scale datasets should be made available via a public repository as described in the *PLOS Genetics*
data availability policy, and numerical data that underlies graphs or summary statistics should be provided in spreadsheet form as supporting information., and numerical data that underlies graphs or summary statistics should be provided in spreadsheet form as supporting information., and numerical data that underlies graphs or summary statistics should be provided in spreadsheet form as supporting information., and numerical data that underlies graphs or summary statistics should be provided in spreadsheet form as supporting information.

Reviewer #1: Yes

Reviewer #2: Yes

PLOS authors have the option to publish the peer review history of their article (what does this mean?). If published, this will include your full peer review and any attached files.). If published, this will include your full peer review and any attached files.). If published, this will include your full peer review and any attached files.). If published, this will include your full peer review and any attached files.

...

Reviewer #1: No

Reviewer #2: No

**Figure resubmission:**
---

## [Editor Report · Decision Letter 2]

2 Apr 2026

Dear Dr Tanaka,

We are pleased to inform you that your manuscript entitled "Constitutively active RAS prolongs Cdc42 signalling, while MAPK signalling is attenuated during fission yeast mating" has been editorially accepted for publication in PLOS Genetics. Congratulations!

Yours sincerely,

Geraldine Butler

Section Editor

PLOS Genetics

Aimée Dudley

Editor-in-Chief

PLOS Genetics

Anne Goriely

Editor-in-Chief

PLOS Genetics

BlueSky: @plos.bsky.social

Comments from the reviewers (if applicable):

**Data Deposition**

If you have submitted a Research Article or Front Matter that has associated data that are not suitable for deposition in a subject-specific public repository (such as GenBank or ArrayExpress), one way to make that data available is to deposit it in the Dryad Digital Repository. As you may recall, we ask all authors to agree to make data available; this is one way to achieve that. A full list of recommended repositories can be found on our . As you may recall, we ask all authors to agree to make data available; this is one way to achieve that. A full list of recommended repositories can be found on our . As you may recall, we ask all authors to agree to make data available; this is one way to achieve that. A full list of recommended repositories can be found on our . As you may recall, we ask all authors to agree to make data available; this is one way to achieve that. A full list of recommended repositories can be found on our website....

http://datadryad.org/submit?journalID=pgenetics&manu=PGENETICS-D-25-00520R2

Additionally, please be aware that our data availability policy requires that all numerical data underlying display items are included with the submission, and you will need to provide this before we can formally accept your manuscript, if not already present. requires that all numerical data underlying display items are included with the submission, and you will need to provide this before we can formally accept your manuscript, if not already present. requires that all numerical data underlying display items are included with the submission, and you will need to provide this before we can formally accept your manuscript, if not already present. requires that all numerical data underlying display items are included with the submission, and you will need to provide this before we can formally accept your manuscript, if not already present.

**Press Queries**

If you or your institution will be preparing press materials for this manuscript, or if you need to know your paper's publication date for media purposes, please inform the journal staff as soon as possible so that your submission can be scheduled accordingly. Your manuscript will remain under a strict press embargo until the publication date and time. This means an early version of your manuscript will not be published ahead of your final version. PLOS Genetics may also choose to issue a press release for your article. If there's anything the journal should know or you'd like more information, please get in touch via plosgenetics@plos.org....

---

## [Editor Report · Acceptance letter]

PGENETICS-D-25-00520R2

Constitutively active RAS prolongs Cdc42 signalling, while MAPK signalling is attenuated during fission yeast mating

Dear Dr Tanaka,

We are pleased to inform you that your manuscript entitled "Constitutively active RAS prolongs Cdc42 signalling, while MAPK signalling is attenuated during fission yeast mating" has been formally accepted for publication in PLOS Genetics! Your manuscript is now with our production department and you will be notified of the publication date in due course.

With kind regards,

Anita Estes

PLOS Genetics

On behalf of:
